# Towards Practical World Model-based Reinforcement Learning for Vision-Language-Action Models

Zhilong Zhang [1 2 *]   Haoxiang Ren [2 *]   Yihao Sun [3 4 *]   Yifei Sheng [1 2]   Haonan Wang [1 2]   Zhichao Wu [1 2]
Haoxin Lin [1 2]   Pierre-Luc Bacon [3 4]   Yang Yu [1 2 †]

## Abstract

Well-pretrained Vision-Language-Action (VLA) models show strong generalization for robotic control, but finetuning them with reinforcement learning (RL) is constrained by the high cost and safety risks of real-world interaction. Training VLA models in interactive world models avoids these issues but introduces several challenges, including pixel-level world modeling, multi-view consistency, and compounding errors under sparse rewards. Building on recent advances across large multimodal models and model-based RL, we propose VLA-MBPO, a practical framework to tackle these problems in VLA finetuning. Our approach has three key design choices: (i) adapting unified multimodal models (UMMs) for data-efficient world modeling; (ii) an interleaved view decoding mechanism to enforce multi-view consistency; and (iii) chunk-level branched rollout to mitigate error compounding. Theoretical analysis and experiments across simulation and real-world tasks demonstrate that VLA-MBPO significantly improves policy performance and sample efficiency, underscoring its robustness and scalability for real-world robotic deployment.

## 1. Introduction

The emergence of Vision-Language-Action (VLA) models represents a significant advancement in robotic control, primarily attributable to their enhanced generalization capabilities (Kim et al., 2024; Intelligence et al., 2025b). By integrating the expansive world knowledge and reasoning ability of Vision-Language Models (VLM) with extensive pretraining on vast robotics datasets, VLA models can effectively learn and execute a diverse range of tasks across varied robotic embodiments (Deng et al., 2025b; Intelligence et al., 2025b; Bjorck et al., 2025).

Previously, finetuning VLA models typically relies on imitation learning (IL), particularly Behavior Cloning (BC), where the model learns a policy by minimizing the discrepancy between the model and expert actions (Ghosh et al., 2024; Kim et al., 2024; 2025; Intelligence et al., 2025b). However, it is well-known that BC is constrained by the limited generalizability and the high cost of collecting large amounts of expert demonstrations (Xu et al., 2020; Rajaraman et al., 2020; Xu et al., 2024). Meanwhile, reinforcement learning (RL) is the standard approach that learns from environmental interactions, thereby achieving stronger generalization with fewer demonstrations (Chen et al., 2025b; Liu et al., 2025; Zang et al., 2025; Li et al., 2025b). However, applying RL in real-world settings remains challenging, as it typically requires numerous interactions with the environment, making it prohibitively expensive, potentially unsafe, and hard to scale (Zhang et al., 2025; Xiao et al., 2025; Zhang et al., 2026).

To enable RL training for VLAs while addressing its limitations in real-world settings, a promising approach is to use world models as simulated environments. This allows agents to learn entirely in imagined spaces, avoiding the high costs and safety risks associated with real-world interaction. Only a few prior studies (Zhang et al., 2024; Zhu et al., 2025) and technical reports (Li et al., 2025a; Fei et al., 2025) have explored this direction, however, applying world models to VLA training still presents several fundamental challenges. First, VLAs rely on visual inputs, requiring world models to generate pixel-level future frames and reward signals that generalize well from limited offline data. While some methods finetune pretrained video models as world models, these models suffer from low inference efficiency and cannot directly predict reward signals (Xiao et al., 2025; Zhu et al., 2025). Second, many robotic tasks, especially fine-grained manipulation, rely on multi-view visual inputs, which demand consistent and coherent pre-

---

*Equal contribution [1]National Key Laboratory for Novel Software Technology, Nanjing University, Nanjing, China [2]School of Artificial Intelligence, Nanjing University, Nanjing, China [3]Mila - Quebec AI Institute [4]Université de Montréal. Correspondence to: Yang Yu <yuy@nju.edu.cn>.

*Proceedings of the 43rd International Conference on Machine Learning*, Seoul, South Korea. PMLR 306, 2026. Copyright 2026 by the author(s).

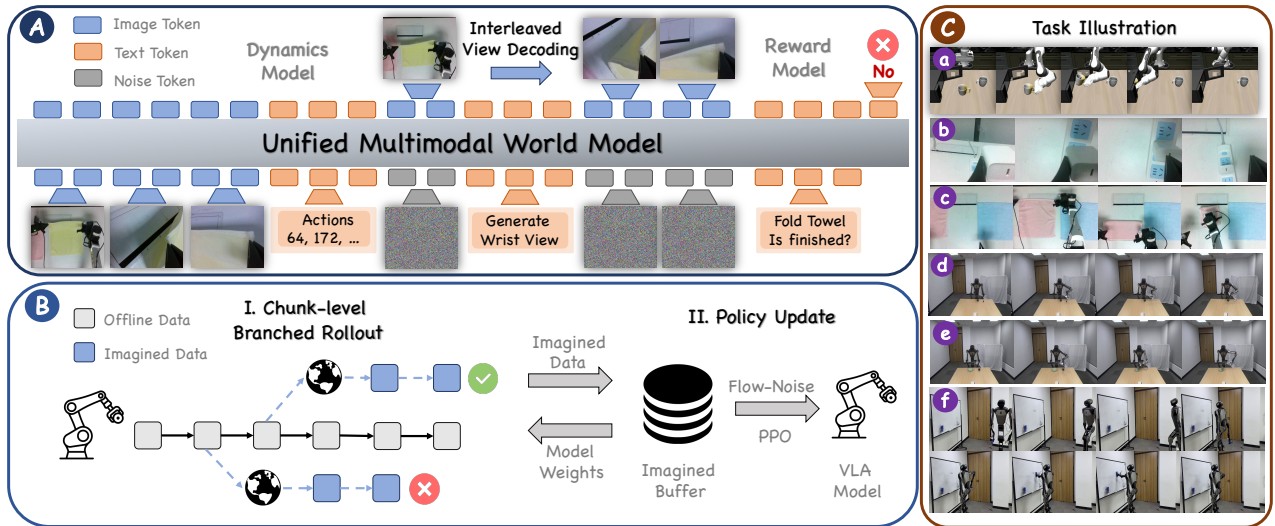

*Figure 1.* The framework illustration of VLA-MBPO, comprising (A) a UMM-based world model with interleaved view decoding for multi-view observation and reward prediction, and (B) a stable and scalable policy update algorithm with chunked-level branch rollout. (C) illustration of both simulated and real-world task designs in our work.

dictions across views (Guo et al., 2025; Qian et al., 2025). Furthermore, model-based learning is inherently susceptible to compounding model errors (Janner et al., 2019; Yu et al., 2020; Xu et al., 2020), and this issue is significantly amplified in VLA settings due to sparse reward signals, where small modeling inaccuracies can lead to opposite reward outcomes and severely misguide policy learning. Together, these challenges make it nontrivial to directly apply existing model-based RL methods to VLA finetuning.

In this work, we propose a practical world model-based RL framework that explicitly addresses these challenges. To enable sample-efficient and generalizable pixel-space world modeling, we use a *pretrained unified multimodal model (UMM)* as the backbone of our world model (Deng et al., 2025a; Cui et al., 2025; Sun et al., 2025), allowing joint prediction of visual dynamics and rewards without expensive video rollouts. To support consistent multi-view generation required for precise control, we introduce the *interleaved view decoding* technique that enforces cross-view consistency while preserving view-specific details. Finally, to mitigate compounding model errors under sparse rewards, we employ *chunk-level branched rollout* that limits error accumulation for policy optimization (Park et al., 2025). Together, these components form a cohesive world model-based RL approach tailored for VLA finetuning (VLA-MBPO), enabling effective reinforcement learning with limited real-world interactions, as illustrated in Figure 1.

To validate VLA-MBPO, we provide a theoretical analysis of the reduced value gap alongside experiments in simulated and real-world tasks. Overall, our results demonstrate that VLA-MBPO substantially improves policy generalization

and data efficiency. Crucially, our approach yields this good performance using a universal set of hyperparameters, paving the way for more reliable and scalable deployment of VLA models in real-world robotic applications.

## 2. Preliminaries

**Language-Conditioned Markov Decision Process.** We formulate the language-conditioned robot control task as an infinite-horizon Language-conditioned Markov Decision Process (L-MDP) $\mathcal{M} = (\mathcal{S}, \mathcal{A}, \mathcal{L}, T, r, \gamma)$ (Puterman, 1990; 2014). The state space $\mathcal{S}$ consists of visual observations, and $\mathcal{L}$ denotes the set of language instructions. The action space $\mathcal{A}$ consists of low-level control commands (e.g., joint positions or end-effector poses). The transition function $T : \mathcal{S} \times \mathcal{A} \rightarrow \Delta(\mathcal{S})$ specifies the distribution over the next state given the current state and the executed action. The reward function $r : \mathcal{S} \times \mathcal{L} \rightarrow \{0, 1\}$ assigns a binary reward based on the visual observation and the task instruction. The objective of L-MDP is to learn a language-conditioned policy $\pi(a_t|s_t, l)$ that maximizes the expected discounted return $V(\pi) = \mathbb{E}_\pi[\sum_{t=0}^{\infty} \gamma^t r(s_t, l)]$ under any task instruction. Recently, most VLAs employ action chunking technique (Fu et al., 2024), where the policy outputs a sequence of $k$ consecutive actions $\tilde{a}_t = (a_t, a_{t+1}, \cdots, a_{t+k-1})$.

**Proximal Policy Optimization.** We optimize the policy $\pi_\theta$ using Proximal Policy Optimization (PPO) (Schulman et al., 2017). PPO aims to maximize a clipped surrogate objective $\mathcal{L}(\theta)$ to constrain policy updates and limit the divergence between $\pi_\theta$ and the behavior policy $\pi_{\theta_{old}}$ that collected the

data. This objective function is computed by:

$$\mathcal{L}(\theta) = \mathbb{E}_t \Big[ \min \Big( \rho_t(\theta) \, \hat{A}_t, \; \mathrm{clip} \big( \rho_t(\theta), 1 - \epsilon, 1 + \epsilon \big) \, \hat{A}_t \Big) \Big]$$
(1)

where $\rho_t(\theta) = \frac{\pi_\theta(a_t|s_t,l)}{\pi_{\theta_{old}}(a_t|s_t,l)}$ is the likelihood ratio. To estimate the advantage $\hat{A}_t$, PPO employ Generalized Advantage Estimation (GAE) (Schulman et al., 2015):

$$\hat{A}_t^{\mathrm{GAE}(\gamma,\lambda)} = \sum_{i=0}^{T-t} (\gamma\lambda)^i \mathcal{T}_t^V$$
(2)

$$\text{where} \quad \mathcal{T}_t^V = r(s_{t+1}, l) + \gamma V_\phi(s_{t+1}, l) - V_\phi(s_t, l)$$

Here $V_\phi(s,l)$ is the language-conditioned value function parameterized by $\phi$, $\gamma$ is the discount factor, and $\lambda$ controls the bias–variance trade-off in the advantage estimation.

## 3. Addressing Practical Challenges in World Model-based RL for VLAs

Applying model-based reinforcement learning (MBRL) for VLA models poses several challenges that arise from pixel-level modeling, multi-view generation, and compounding errors under sparse rewards. We organize this section around these challenges and describe how each design choice addresses a specific problem. We first focus on challenges in dynamics and reward modeling in the VLA domain (Section 3.1), and then tackle compounding errors exacerbated by sparse rewards during policy learning (Section 3.2). Together, these components form a cohesive and practical world model-based RL framework for VLA (Section 3.3).

### 3.1. Challenges in World Modeling for VLAs

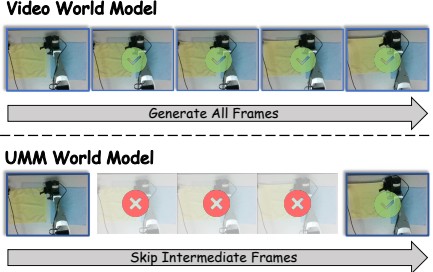

*Figure 2.* Frame-skipping scheme in UMM-based World Model.

Unlike conventional MBRL settings that rely on low-dimensional states or latent rollouts (Janner et al., 2019; Hafner et al., 2023; Hansen et al., 2024), world models for VLAs require (i) high-fidelity pixel-based generation since VLAs typically take origin images as inputs for visual perception, (ii) consistent multi-view generation for fine-grained control, and (iii) accurate semantic understanding ability for reward modeling. However, training such pixel-space world models from scratch is notoriously data-hungry and prone to overfitting in offline settings.

Prior work typically finetunes separate large video models and vision–language models for dynamics and reward modeling, respectively (Zhu et al., 2025). While effective, this two-model design introduces complexity and engineering overhead. In contrast, inspired by recent advances (Deng et al., 2025a; Sun et al., 2025), we adopt unified multimodal models (UMMs) as a streamlined alternative, enabling joint prediction of future observations and rewards within a single model. Moreover, by directly modeling dynamics without generating intermediate frames, UMMs improve roll-out efficiency compared to video world models (Figure 2). However, since UMMs are not originally tailored for VLA settings, additional adaptations are required.

**Extending UMMs to Action Spaces.** UMM-based world models typically operate over vision and language modalities (Sun et al., 2025), whereas VLA agents introduce an additional low-level action modality. To support UMMs in handling low-level action inputs, we follow Goyal et al. (2025) to represent actions as integer tokens by discretizing continuous action values into a fixed range (e.g., [0,256]) and mapping them to the UMM vocabulary. The UMM is then tasked with generating the next observation and reward based on the text-based action chunks. This process is formalized by the conditional probability $s_{t+k} \sim T_\theta(\cdot | s_t, \tilde{a}_t)$ for the next observation, where $\tilde{a}_t$ represents a $k \times d$ token sequence, with $k$ being the chunk size and $d$ the number of action dimensions. Moreover, we define the chunk-level reward as $r_\theta(s_{t+k}, l) = \sum_{i=1}^{k} \gamma^{i-1} r(s_{t+i}, l)$, where $\gamma$ is the discount factor and $r(s_{t+i}, l)$ denotes the reward at step $t+i$ under task instruction $l$. This design requires no architectural or vocabulary modification, preserving the pretrained capabilities of UMMs. In our implementation, we adopt Bagel (Deng et al., 2025a) as the base model.

**Interleaved View Decoding.** Fine-grained manipulation often requires reasoning over multiple camera views, as a single viewpoint is insufficient to fully capture object geometry, occlusions, and contact dynamics. This introduces an additional challenge for world modeling: in addition to predicting visually plausible future observations, the world model must maintain cross-view consistency for downstream policy learning. Directly extending UMMs to multi-view inputs often results in view-specific artifacts, which can degrade control performance, even when local predictions are accurate. To address this challenge, we propose a *interleaved view decoding* strategy that explicitly enforces consistency across multiple camera views. In most VLA models, the input consists of both head-view (or top-view) camera images $s^h$ and wrist-camera images $s^w$, forming the combined input $s = [s^h, s^w]$. Here, the head-view captures global scene information, while the wrist-view provides fine-grained but partially observable details. To model this, we decompose

the state transitions as:

$$
\begin{aligned}
s_{t+k}^h &\sim T_\theta(\cdot | s_t^h, s_t^w, a_{t:t+k-1}) \\
s_{t+k}^w &\sim T_\theta(\cdot | s_t^w, s_{t+k}^h)
\end{aligned}
\tag{3}
$$

Empirically, we find that this way is better than independent generation of each view in Section 5.1, which ensures that both global and fine-grained information are effectively integrated, maintaining consistency between views. This decomposition can be implemented easily with interleaved decoding in UMMs with attention matrix in Figure 10.

### 3.2. Challenges in Compounding Model Errors under Sparse Rewards

Compounding errors are a fundamental challenge in MBRL, as inaccuracies in world model predictions accumulate over long rollouts and can severely mislead policy optimization. In VLA settings, this issue is further exacerbated by sparse reward structures commonly encountered in manipulation tasks, where small prediction errors may result in qualitatively different outcomes and even opposite reward signals. Such error amplification renders naive full-horizon rollout strategies unreliable.

To mitigate this issue, we employ the *chunk-level branched rollout*, a technique used in state-based simple tasks (Park et al., 2025) but has not been validated in pixel-based VLA finetuning. Instead of rolling out full-horizon trajectories starting from the initial state $s_0$, we initiate rollouts from any observation within the offline dataset with much smaller rollout horizon. Furthermore, since our world model operates at the chunk level, we can further reduce the effective rollout horizon by a factor of $1/k$, where $k$ is the chunk size. By combining these two strategies, we shorten the rollout length by a large margin, thereby enhancing the efficiency and stability of policy optimization.

### 3.3. VLA-MBPO: Practical World Model-based RL Framework for VLA Models

We present **VLA-MBPO**, a practical world model-based RL framework for VLA models, which integrates the three components mentioned above to facilitate VLA reinforcement learning, as illustrated in Figure 1. Our algorithm consists of three phases: 1) data collection with the VLA model; 2) world model finetuning with collected data; 3) policy optimization with RL in the world model. For policy optimization, we adopt *Flow-Noise* (Chen et al., 2025a), a simple variant of PPO for flow matching-based policy learning. During the RL process, we append an MLP-based value head to the VLA model for value prediction (Chen et al., 2025a). Here, the advantage under $n$ branched rollout

with chunk size $k$ is defined as:

$$
\hat{A}_t^{\mathrm{GAE}(\gamma,\lambda)} = \sum_{i=0}^{n} (\gamma^k \lambda)^i \mathcal{T}_t^V , \text{ where}
$$

$$
\mathcal{T}_t^V = \sum_{j=1}^{k} \gamma^{j-1} r(s_{t+j}, l) + \gamma^k V_\phi(s_{t+k}, l) - V_\phi(s_t, l)
\tag{4}
$$

The Flow-Noise estimates the log likelihood term in the PPO policy loss as follows:

$$
\begin{aligned}
&\log \pi_\phi(\mathcal{A} | s_t) \\
&= \log \left( \pi_\phi\left(\mathbf{A}^0 | s_t\right) \prod_{i=0}^{K-1} \pi_\phi\left(\mathbf{A}^{\tau_{i+1}} | \mathbf{A}^{\tau_i}, s_t\right) \right)
\end{aligned}
\tag{5}
$$

where $\mathcal{A} = (\mathbf{A}^0, ..., \mathbf{A}^1)$ is denoising sequence of an action chunk. The complete algorithm is shown in Algorithm 1.

---

**Algorithm 1** VLA-MBPO

---

1: **Require:** Initialized VLA Model $\{\pi_\phi, V_\phi\}$, initialized World Model $\{T_\theta, r_\theta\}$, initialized the replay buffer $\mathcal{D}$.
2: **Data collection:** Run $\pi_\phi$ in real environments to collect data and add them to $\mathcal{D}$.
3: **World model fine-tuning:** Fine-tune $\{T_\theta, r_\theta\}$ on $\mathcal{D}$.
4: **Policy optimization in WM:**
5: **for** $j = 1$ **to** $N_{\mathrm{RL\_iter}}$ **do**
6:     Sample starting states $\{s_t\}^M$ from buffer $\mathcal{D}$.
7:     Generate chunk-level branched rollouts by $T_\theta, r_\theta$.
8:     Run *Flow-Noise* to update $\{\pi_\phi, V_\phi\}$ on these synthetic data.
9: **end for**

---

Our method can be viewed as an instance of offline model-based reinforcement learning (MBRL), but it differs from prior offline MBRL approaches in several key aspects. First, unlike traditional methods using conservative regularization to mitigate model bias (Yu et al., 2020; Sun et al., 2023; Lin et al., 2025), our approach omits such mechanisms as the finetuned UMM-World achieves sufficient accuracy to render them unnecessary. Second, unlike recent action-chunking-based offline MBRL method (Park et al., 2025), our method is built upon PPO framework and therefore does not rely on additional designs like rejection sampling and Q models, significantly reducing the system complexity. Driven by these two advantages, **our method maintains a single set of hyperparameters across all tasks** (Table 5), which enhances its practical utility and simplifies deployment in real-world scenarios.

## 4. Theoretical Analysis

In this section, we first analyze how previous world model-based RL methods suffer from severe compounding model error in Section 4.1. Then we show a theoretical result of how VLA-MBPO mitigates these errors in Section 4.2.

### 4.1. Value Gap of World Model-based RL

Previous world model-based RL for VLA methods evaluate the chunk-level policy in the step-level world model with full-horizon rollouts (Xiao et al., 2025; Zhu et al., 2025).

**Theorem 4.1** (Value Gap of Chunk-level Policy with Step-level World Model). *Given a chunk-level policy $\pi^k$ with a chunk size $k$, and a step-level world model $\hat{T}$. Assume that $\epsilon_\pi^k = \max_s D_{\text{TV}}\left(\pi_{\mathcal{D}}^k(\tilde{a}|s)\|\pi^k(\tilde{a}|s)\right)$ and $\epsilon_m = \max_t \mathbb{E}_{s\sim\mathcal{D}^t}\left[D_{\text{TV}}\left(T(s'|s,a)\|\hat{T}(s'|s,a)\right)\right]$. Then, the value gap of policy $\pi^k$ under the true environment and the learned world model is bounded by:*

$$
\begin{aligned}
&|V(\pi^k) - \hat{V}(\pi^k)| \\
&\leq \frac{2r_{\max}}{1-\gamma}\left[\frac{2\gamma^k}{1-\gamma^k}\epsilon_\pi^k + 2\epsilon_\pi^k + \frac{k\gamma^k}{1-\gamma^k}\epsilon_m\right]
\end{aligned}
\tag{6}
$$

*Proof.* See Appendix B. ∎

As shown in Theorem 4.1, the value gap is influenced by two components: (a) **Policy error** $\epsilon_\pi^k$: the divergence between the current policy $\pi^k$ and the behavior policy $\pi_{\mathcal{D}}^k$, with dependency $\mathcal{O}\left(\frac{\gamma^k}{(1-\gamma)(1-\gamma^k)}\right)$. (b) **Model error** $\epsilon_m$: the discrepancy between true dynamics $T$ and learned model $\hat{T}$, which also significantly scales as $\mathcal{O}\left(\frac{k\gamma^k}{(1-\gamma)(1-\gamma^k)}\right)$.

The value gap grows quadratically with the task horizon due to the compounding errors, resulting in big value errors in long-horizon tasks. For large VLA models, we typically control policy divergence $\epsilon_\pi^k$ by reducing the update step size or adding a KL constraint. However, world model error $\epsilon_m$ is more challenging to control during RL training, as it is largely dependent on the high-quality pretraining. Consequently, even small model inaccuracies can accumulate over long rollouts, leading to significant distortions in value estimates and misleading policy optimization.

### 4.2. Reduced Value Gap of VLA-MBPO

Here, we demonstrate how VLA-MBPO reduces compounding errors by employing a chunk-level branched rollouts in a chunk-wise world model.

**Theorem 4.2** (Value Gap of Chunk-level Policy with Chunk-level World Model and Branched Rollouts). *Given an offline dataset $\mathcal{D}$ and a chunk-level policy $\pi^k$ with a chunk size $k$. Suppose that policy $\pi^k$ is evaluated via $n$-chunks branched rollouts in the chunk-level world model $\hat{T}^k(s'|s,\tilde{a})$. Assume that $\epsilon_\pi^k = \max_s D_{\text{TV}}\left(\pi_{\mathcal{D}}^k(\tilde{a}|s)\|\pi^k(\tilde{a}|s)\right)$ and $\epsilon_m^{k,n} = \max_{t\leq n}\mathbb{E}_{s\sim d_{t,s_0}^{\pi^k}\sim\mathcal{D}}\left[D_{\text{TV}}\left(T^k(s'|s,\tilde{a})\|\hat{T}^k(s'|s,\tilde{a})\right)\right]$. Then the value gap of policy $\pi^k$ under the true environment*

*and the learned world model is bounded by:*

$$
\begin{aligned}
&\left|V(\pi^k) - \hat{V}^{branch}(\pi^k)\right| \\
&\leq \frac{2r_{\max}}{1-\gamma}\left[\frac{(\gamma^k)^{n+1}}{1-\gamma^k}\epsilon_\pi^k + (\gamma^k)^n\epsilon_\pi^k + n\epsilon_m^{k,n}\right]
\end{aligned}
\tag{7}
$$

*Proof.* See Appendix B. ∎

Theorem 4.2 highlights how VLA-MBPO limits the value gap between the true environment and the learned world model via chunk-level world models and branched rollouts. By restricting the length of the rollouts and breaking the task into chunks, both the policy and model errors are mitigated compared to previous methods. Specifically, the policy error dependency is bounded by $\mathcal{O}\left(\frac{(\gamma^k)^{n+1}}{(1-\gamma)(1-\gamma^k)}\right)$ while the model error dependency is bounded by $\mathcal{O}\left(\frac{n}{1-\gamma}\right)$. We observe that as $n$ increases, while the dependency on model error increases linearly, the influence of policy error diminishes exponentially. This suggests that by appropriately selecting the branched-rollout chunks $n$, we can significantly alleviate this value gap.

**Simple Case Study.** Consider a scenario where the discount factor $\gamma = 0.99$ and the chunk size $k = 10$, which are typical choices in VLA-RL tasks. In such cases, previous world model-based VLA-RL methods suffer from a value gap of approximately $4183\epsilon_\pi^k + 18916\epsilon_m$. In contrast, if we use VLA-MBPO with branched rollouts of length $n = 2$, the value gap is considerably smaller, approximately $1710\epsilon_\pi^k + 400\epsilon_m^{k,n}$, demonstrating the effectiveness of the VLA-MBPO in mitigating error compounding.

## 5. Experiments

We conduct extensive experiments on both simulated and real-world robotic tasks to answer these questions. **Q1**: How does our UMM-based world model (UMM-World) perform on multi-view dynamics modeling and reward prediction? (Section 5.1) **Q2**: How does VLA-MBPO perform in simulation tasks? (Section 5.2) **Q3**: How does VLA-MBPO perform in real-world tasks? (Section 5.3) **Q4**: How sensitive is VLA-MBPO to key hyperparameters and design choices? (Section 5.4)

### 5.1. World Model Evaluation

**Benchmark.** We conduct our evaluation on the *Object* task suite in LIBERO (Liu et al., 2023), a manipulation benchmark that contains 10 distinct tasks with diverse object instances. We use a dataset comprised of 50 trajectories per task for training, and report evaluation results on a held-out test set of 10 trajectories per task. The evaluation protocol involves rolling out 40 steps across 100 held-out test trajec-

*Table 1.* Performance comparison of various world models across multiple metrics, including dynamics and reward model evaluations.

| Model | Dynamics Model | | | | | | | Reward Model | |
|---|---|---|---|---|---|---|---|---|---|
| | Head View | | | Wrist View | | | Inf. Time↓ | ACC↑ | F1↑ |
| | LPIPS↓ | PSNR↑ | SSIM↑ | LPIPS↓ | PSNR↑ | SSIM↑ | | | |
| Ctrl-World | 0.150 | 21.95 | 0.882 | 0.435 | 13.87 | 0.680 | 21 | – | – |
| Qwen3-VL-8B | – | – | – | – | – | – | – | 97.0 | 0.841 |
| **UMM-World** | **0.094** | **23.29** | **0.906** | **0.254** | **18.76** | **0.751** | 10 | 98.4 | **0.861** |
| *- w/o IVD* | 0.116 | 21.71 | 0.895 | 0.454 | 13.38 | 0.559 | **8** | **98.5** | 0.799 |
| *- w/o PT* | 0.281 | 19.26 | 0.756 | 0.579 | 12.80 | 0.499 | 10 | 94.5 | 0.496 |

tories. We report evaluation results for both head and wrist views to rigorously assess long-term consistency.

**Baselines.** We quantitatively compare our model against two distinct baselines: (1) *Ctrl-World* (Guo et al., 2025), a video generation model that excels at dynamics synthesis with multi-view consistency but lacks the semantic grounding for intrinsic reward prediction; and (2) *Qwen3-VL* (Bai et al., 2025), a VLM capable of precise reward reasoning but unable to predict visual dynamics. Moreover, to validate our model design, we evaluate two ablations: 1) *w/o Interleaved View Decoding (IVD)*, where views are generated in parallel rather than interleaved, isolating the impact of our decoding strategy on multi-view consistency. 2) *w/o Pretrained (PT)*, where UMM-World is randomly initialized. We quantitatively evaluate all models from dynamics prediction, inference speed and reward prediction perspectives.

**Task Results.** As presented in Table 1, UMM-World demonstrates superior performance across nearly all metrics. Not only does it outperforms the video world model baseline on the prediction fidelity of both head and wrist views, but also achieves a $2\times$ faster inference speed owing to the frame-skipping scheme. When it comes to reward prediction, our model also matches the specialist Qwen3-VL-8B. We credit these substantial gains to two key innovations: *Interleaved View Decoding* and *Unified Pretraining*. Ablation studies confirm that removing either component leads to a severe degradation in both dynamics modeling and reward prediction performance. We also present some qualitative results of UMM-World in Figure 4 and Figure 11.

### 5.2. Simulation Task Experiments

**Benchmark.** We evaluate VLA-MBPO on LIBERO (Liu et al., 2023), a widely adopted benchmark consisting of four task suites: *Spatial*, *Object*, *Goal*, and *Long*, designed to assess capabilities ranging from visual grounding to long-horizon planning. The offline dataset is constructed by collecting 50 episodes per task, where the behavior policy is obtained via $\pi_{0.5}$ with one-shot SFT. We measure performance by average success rate over 50 evaluation episodes per task across all 10 tasks in each suite.

**Baselines.** We compare VLA-MBPO against four baselines: (1) $\pi_{0.5}$ (SFT): the VLA model ($\pi_{0.5}$) prior to any RL; (2) *BC* (WM): a BC baseline trained on successful trajectories generated by world models; (3) an online RL baseline $\pi_{RL}$ (Chen et al., 2025a) under an equivalent real-world interaction budget; and (4) *IDQL* (Hansen-Estruch et al., 2023): an offline model-free RL algorithm for flow-matching policies.

*Table 2.* Performance comparison on LIBERO benchmark.

| Model | LIBERO | | | | |
|---|---|---|---|---|---|
| | Spatial | Object | Goal | Long | Avg |
| One-Trajectory SFT | | | | | |
| $\pi_{0.5}$ (SFT) | 78.2 | 88.6 | 85.8 | 54.6 | 76.8 |
| BC (WM) | 80.6 | 89.8 | 85.0 | 48.6 | 76.0 |
| $\pi_{RL}$ | 86.0 | 92.4 | 90.8 | 61.2 | 82.6 |
| IDQL | 79.0 | 92.4 | 86.4 | 52.2 | 77.5 |
| **VLA-MBPO** | **87.8** | **96.6** | **92.8** | **66.8** | **85.9** |
| Δ | +9.6 | +8.0 | +6.8 | +12.2 | +9.1 |

**Task Results.** As shown in Table 2, VLA-MBPO achieves consistent and substantial performance gains over all baselines across every suite of the LIBERO benchmark. It attains the highest average success rate, improving the initial SFT policy by +9.1. Notably, the most pronounced improvement occurs on the *LIBERO-Long* suite, which comprises the most challenging long-horizon tasks. Under an identical real-world data budget, VLA-MBPO substantially outperforms both offline and online RL methods on this suite, highlighting its superior sample efficiency and effectiveness in solving complex manipulation tasks.

**Generalizable Credit Assignment.** We visualize the learning dynamics of the value model on unseen full-horizon trajectories in Figure 3. We find that despite training with short branched rollouts, our value model progressively aligns with ground-truth returns through learning cross-chunk temporal dependencies and value consistency, thereby achieving coherent long-horizon value estimates that accurately reflect trajectory outcomes. We posit that this stitching capability is the fundamental mechanism enabling VLA-MBPO

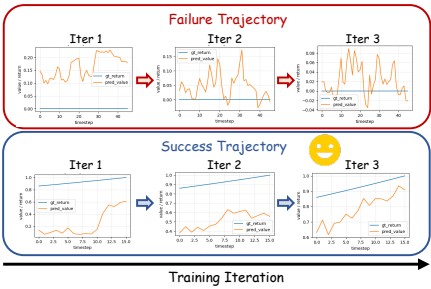

*Figure 3.* The learning dynamics of value models. The blue lines represent the ground-truth return, while the orange lines represent the return estimated by the value model.

to achieve significant policy improvement using only short-horizon model-based rollouts.

### 5.3. Real-world Task Experiments

Transitioning from simulation to the physical world introduces significant challenges, including complex non-rigid dynamics, sensor noise, and unmodeled environmental factors. In this section, we explore whether VLA-MBPO can improve performance of VLAs in the real-world scenarios.

**Experiment Setup.** As shown in Figure 9, we design five real-world tasks across two robotic platforms. On the bimanual robot **Arx-X5**, a) *Plug Cable* requires sub-centimeter precision to insert a cable into a 3-mm socket, while b) *Fold Towel* assesses bimanual manipulation of deformable objects. On the whole-body robot **Galaxy-R1**, c) *Pick Cup* and d) *Insert Pen* evaluate whole-body manipulation with disturbed robot poses and camera viewpoints, and e) *Wipe Board* tests mobile whole-body control under partial observability. For each task, we collect expert demonstrations via human teleoperation, with approximately 50 trajectories for Arx-X5 tasks and 100 trajectories for Galaxy-R1 tasks, and perform SFT for $\pi_{0.5}$. We then collect 50 rollouts per task using $\pi_{0.5}$ (SFT) for subsequent VLA-MBPO training. Evaluation is conducted on 50 trajectories per task: 30 *seen* and 20 *unseen*, where the latter include novel objects, backgrounds, and spatial configurations. Additional details are provided in Appendix C.

**Task Results.** As shown in Figure 5, VLA-MBPO delivers consistent performance gains across both robotic platforms, validating its robustness in physical environments where dynamics are inherently stochastic. On Arx-X5, gains in *Fold Towel* confirm effective modeling of deformable-object dynamics, while improved *Plug Cable* performance highlights success in fine-grained, contact-rich manipulation. On Galaxy-R1, VLA-MBPO excels in high-DoF whole-body control, notably in *Wipe Board* task with severe partial observability. Strong results on both seen and unseen conditions further validate its real-world generalization.

### 5.4. Ablation Study

While the main results highlight the effectiveness of VLA-MBPO, we now explore the influence of two key factors: the a) *rollouts scheme*, and b) *generated sample size*. We conduct all ablation studies on *LIBERO-Long* suite.

*Table 3.* Results of different rollouts scheme for VLA-MBPO.

| Rollouts Scheme | LIBERO Long | | | |
| | Branched Rollout | | | Full Horizon |
| | 1 | 2 | 4 | |
|---|---|---|---|---|
| **VLA-MBPO** | 63.9 | **66.8** | 62.9 | 52.8 |

**Rollouts Scheme.** We analyze the impact of different rollout lengths for VLA-MBPO, as shown in Table 3. The results consistently demonstrate that VLA-MBPO improves policy performance across various rollout horizons. However, there is a tradeoff: shorter rollouts (length=1 chunk) restrict exploration and complicate trajectory stitching, while excessively long horizons (length=4 chunks) lead to the unreliable trajectory generation. Both of these factors limit the potential for performance gains. Furthermore, using full-horizon rollouts in long-horizon tasks results in significant performance degradation due to intolerable compounding model errors.

**Imagined Sample Size.** We analyze the policy performance across different generated sample sizes per iteration as shown in Figure 6. VLA-MBPO demonstrates a consistently monotonic improvement in success rate with increasing sample size. As the generated sample size grows, the value estimations become more accurate, which reduces biases in policy gradients and results in more stable policy improvements. This underscores the potential of VLA-MBPO to tackle more complex and diverse multitask learning challenges, offering a scalable and practical approach for world model-based RL for VLAs.

## 6. Related Works

**Reinforcement Learning for VLA Models.** Recent research has increasingly focused on adapting reinforcement learning to fine-tune VLA models for embodied control. Li et al. (2025b) and Liu et al. (2025) adapt standard on-policy algorithms (Schulman et al., 2017) to fine-tune next-token prediction VLA models (Kim et al., 2024; 2025) and other work like $\pi_{RL}$ (Chen et al., 2025a) extends online fine-tuning to flow matching policies (Black et al., 2024; Intelligence et al., 2025c). However, regardless of the backbone, the prohibitively high sample complexity of online interaction restricts their applicability in real-world scenarios. To circumvent these costs, Offline RL has been applied to fine-tuning VLA. $\pi_{0.6}^*$ (Intelligence et al., 2025a) improves VLA policies via advantage-conditioned policy optimization over

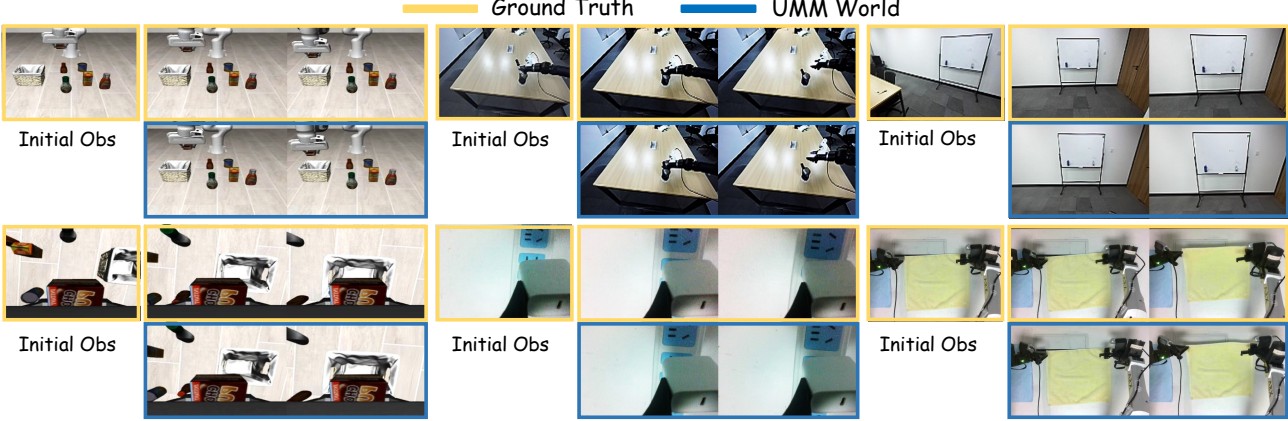

*Figure 4.* Qualitative visualization results of UMM-World.

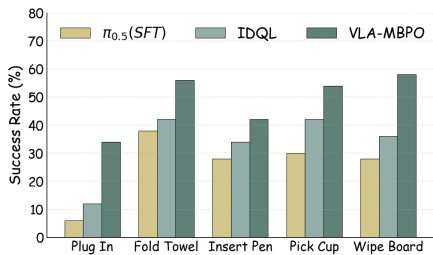

*Figure 5.* Performance comparison on real-world tasks.

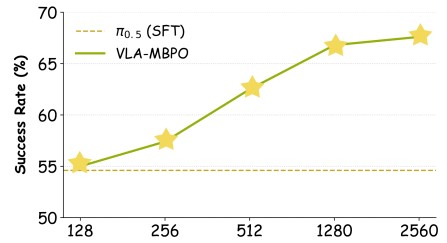

*Figure 6.* Results of different sample size for VLA-MBPO.

heterogeneous offline data. Reinbot (Zhang et al., 2025) enables offline policy optimization by predicting dense returns from static datasets. Despite their data efficiency, these methods operate purely on static datasets without modeling environment dynamics, limiting their ability to simulate consequences or recover from out-of-distribution states.

**Model-based Reinforcement Learning.** Model-based reinforcement learning improves sample efficiency by learning environment dynamics to augment experience with imagined transitions (Janner et al., 2019; Lin et al., 2023) or to enable planning (Chua et al., 2018). This paradigm has also been extended to offline RL by generating synthetic data beyond static datasets (Yu et al., 2020; Sun et al., 2023; Lin et al., 2025; 2026; Wu et al., 2026). A central challenge in MBRL is *compounding error*, where small prediction errors accumulate over rollout horizons. MBPO (Janner et al., 2019) mitigates error by performing short branched

rollouts, and MOPO (Yu et al., 2020) and MOBILE (Sun et al., 2023) further penalize policy optimization in regions with high uncertainty to prevent exploitation of inaccurate dynamics. More recently, Park et al. (2025) propose an action-chunked dynamics model that predicts future states from action sequences, though it remains limited to low-dimensional state spaces and relies on a complex design for conservative Q-learning. In high-dimensional visual control and VLA settings, Zhu et al. (2025); Xiao et al. (2025); Li et al. (2025a) make important progress, but their reliance on full-horizon rollouts leaves them vulnerable to compounding errors, limiting their applicability to long-horizon and complex tasks.

## 7. Conclusion

In this paper, we present VLA-MBPO, a practical world model-based RL framework for training VLA models. By integrating a UMM-based backbone, interleaved view decoding for multi-view consistency, and chunk-level branched rollout to mitigate error compounding, VLA-MBPO significantly enhances policy improvement and sample efficiency. These advancements enables a more practical and scalable post-training method for VLA models, pave a way for more reliable real-world deployment.

**Limitations.** Despite the higher inference efficiency of UMM-World compared to video models, the sample generation phase still requires large computational resources, as detailed in Appendix F. Additionally, since the UMM model used in our framework was not pretrained on action-labeled robotic data, VLA-MBPO still requires a small amount data to finetune the world model when applied to downstream tasks. A promising direction for future work is to enhance the pretraining process, enabling the world model to achieve one-shot or even zero-shot generalization.

## Impact Statement

This paper presents work whose goal is to advance the field of Machine Learning. There are many potential societal consequences of our work, none of which we feel must be specifically highlighted here.

## Acknowledgements

This work was supported by the Yangtze River Delta Science and Technology Innovation Community Joint Research Program 2024CSJZN00302, the National Natural Science Foundation of China under Grants 62495090 and 62495093, the Natural Science Foundation of Jiangsu under Grants BK20243039, the "111 Center" (No. B26023), and the Fundamental and Interdisciplinary Disciplines Breakthrough Plan of the Ministry of Education of China (No. JYB2025XDXM118).

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

## A. Useful Lemmas

**Lemma A.1** (Total Variation Distance for Joint Distributions). *Let $\mathbb{P}_1(x, y) = \mathbb{P}_1(x)\,\mathbb{P}_1(y|x)$ and $\mathbb{P}_2(x, y) = \mathbb{P}_2(x)\,\mathbb{P}_2(y|x)$ be two joint probability distributions over $(x, y)$. Then the total variation distance between $\mathbb{P}_1$ and $\mathbb{P}_2$ satisfies*

$$D_{\mathrm{TV}}\big(\mathbb{P}_1(x, y)\|\mathbb{P}_2(x, y)\big) \leq D_{\mathrm{TV}}\big(\mathbb{P}_1(x)\|\mathbb{P}_2(x)\big) + \max_x D_{\mathrm{TV}}\big(\mathbb{P}_1(y|x)\|\mathbb{P}_2(y|x)\big) \tag{8}$$

*Proof.* Recall that the total variation distance between two distributions $\mu$ and $\nu$ is defined as $D_{\mathrm{TV}}(\mu, \nu) = \frac{1}{2}\int |d\mu - d\nu|$. Applying this to the joint distributions $\mathbb{P}_1$ and $\mathbb{P}_2$, we have

$$
\begin{aligned}
&D_{\mathrm{TV}}\big(\mathbb{P}_1(x, y)\|\mathbb{P}_2(x, y)\big) \\
&= \frac{1}{2}\int_{x,y} \big|\mathbb{P}_1(x)\mathbb{P}_1(y|x) - \mathbb{P}_2(x)\mathbb{P}_2(y|x)\big|\, dx\, dy \\
&= \frac{1}{2}\int_{x,y} \big|\mathbb{P}_1(x)\mathbb{P}_1(y|x) - \mathbb{P}_1(x)\mathbb{P}_2(y|x) + \mathbb{P}_1(x)\mathbb{P}_2(y|x) - \mathbb{P}_2(x)\mathbb{P}_2(y|x)\big|\, dx\, dy \\
&\leq \frac{1}{2}\int_{x,y} \mathbb{P}_1(x)\big|\mathbb{P}_1(y|x) - \mathbb{P}_2(y|x)\big|\, dx\, dy + \frac{1}{2}\int_{x,y} \big|\mathbb{P}_1(x) - \mathbb{P}_2(x)\big|\mathbb{P}_2(y|x)\, dx\, dy \\
&= \mathbb{E}_{x\sim\mathbb{P}_1}\big[D_{\mathrm{TV}}\big(\mathbb{P}_1(y|x)\|\mathbb{P}_2(y|x)\big)\big] + D_{\mathrm{TV}}\big(\mathbb{P}_1(x)\|\mathbb{P}_2(x)\big) \\
&\leq \max_x D_{\mathrm{TV}}\big(\mathbb{P}_1(y|x)\|\mathbb{P}_2(y|x)\big) + D_{\mathrm{TV}}\big(\mathbb{P}_1(x)\|\mathbb{P}_2(x)\big)
\end{aligned}
$$

$\square$

**Lemma A.2** (Total Variation Bound for Rollout State Distributions). *Let $\mathbb{P}_1$ and $\mathbb{P}_2$ be two stochastic processes over states $\{s_t\}_{t\geq 0}$ that share the same initial state distribution, i.e., $\mathbb{P}_1(s_0) = \mathbb{P}_2(s_0)$. Suppose that the expected total variation distance between their transition kernels satisfies $\max_t \mathbb{E}_{s\sim\mathbb{P}_1^t(s)}\big[D_{\mathrm{TV}}\big(\mathbb{P}_1(s'|s)\|\mathbb{P}_2(s'|s)\big)\big] \leq \delta$. Then, the total variation distance between the marginal state distributions at time $t$ is bounded as*

$$D_{\mathrm{TV}}\big(\mathbb{P}_1(s_t)\|\mathbb{P}_2(s_t)\big) \leq t\delta \tag{9}$$

*Proof.* We proceed by induction on $t$. For $t = 0$, we have $\mathbb{P}_1(s_0) = \mathbb{P}_2(s_0)$ by assumption, so $D_{\mathrm{TV}}(\mathbb{P}_1(s_0)\|\mathbb{P}_2(s_0)) = 0 \leq 0 \cdot \delta$, which establishes the base case.

Assume the claim holds for some $t - 1 \geq 0$, i.e.,

$$D_{\mathrm{TV}}\big(\mathbb{P}_1(s_{t-1})\|\mathbb{P}_2(s_{t-1})\big) \leq (t-1)\delta \tag{10}$$

We now bound the distance at step $n$. By the law of total probability, the marginal densities satisfy

$$\mathbb{P}_i(s_t) = \int \mathbb{P}_i(s_t|s_{t-1})\,\mathbb{P}_i(s_{t-1})\, ds_{t-1}, \quad i \in \{1, 2\}$$

The total variation distance is then

$$
\begin{aligned}
&D_{\mathrm{TV}}\big(\mathbb{P}_1(s_t),\, \mathbb{P}_2(s_t)\big) \\
&= \frac{1}{2}\int \big|\mathbb{P}_1(s_t) - \mathbb{P}_2(s_t)\big|\, ds_t \\
&= \frac{1}{2}\int \left|\int \Big[\mathbb{P}_1(s_t|s_{t-1})\mathbb{P}_1(s_{t-1}) - \mathbb{P}_2(s_t|s_{t-1})\mathbb{P}_2(s_{t-1})\Big] ds_{t-1}\right| ds_t \\
&\leq \frac{1}{2}\int\int \big|\mathbb{P}_1(s_t|s_{t-1})\mathbb{P}_1(s_{t-1}) - \mathbb{P}_2(s_t|s_{t-1})\mathbb{P}_2(s_{t-1})\big| ds_{t-1}\, ds_t \\
&\leq \frac{1}{2}\int\int \mathbb{P}_1(s_{t-1})\big|\mathbb{P}_1(s_t|s_{t-1}) - \mathbb{P}_2(s_t|s_{t-1})\big|\, ds_t\, ds_{t-1} + \frac{1}{2}\int\int \big|\mathbb{P}_1(s_{t-1}) - \mathbb{P}_2(s_{t-1})\big|\, \mathbb{P}_2(s_t|s_{t-1})\, ds_t\, ds_{t-1} \\
&= \frac{1}{2}\int\int \mathbb{P}_1(s_{t-1})\big|\mathbb{P}_1(s_t|s_{t-1}) - \mathbb{P}_2(s_t|s_{t-1})\big|\, ds_t\, ds_{t-1} + \frac{1}{2}\int \big|\mathbb{P}_1(s_{t-1}) - \mathbb{P}_2(s_{t-1})\big|\, ds_{t-1} \\
&= \mathbb{E}_{s_{t-1}\sim\mathbb{P}_1^{t-1}(s)}\big[D_{\mathrm{TV}}\big(\mathbb{P}_1(s_t|s_{t-1})\|\mathbb{P}_2(s_t|s_{t-1})\big)\big] + D_{\mathrm{TV}}\big(\mathbb{P}_1(s_{t-1})\|\mathbb{P}_2(s_{t-1})\big)
\end{aligned}
$$

$$\tag{11}$$

By the assumption, the first term is at most $\delta$, and by the induction hypothesis, the second term is at most $(n-1)\delta$. Therefore,

$$D_{\text{TV}}\big(\mathbb{P}_1(s_t)\|\mathbb{P}_2(s_t)\big) \leq \delta + (n-1)\delta = n\delta \tag{12}$$

This completes the induction step. Hence, for all $n \geq 0$,

$$D_{\text{TV}}\big(\mathbb{P}_1(s_t)\|\mathbb{P}_2(s_t)\big) \leq n\delta \tag{13}$$

$\square$

**Lemma A.3** (Value Divergence, Chunk-Level Policy, Chunk-Level World Model). *Let $k \in \mathbb{N}^+$ be the output chunk size of policies. Consider two stochastic $k$-steps policies $\pi_1^k$ and $\pi_2^k$, where each policy selects an action sequence $\tilde{a}_t = a_{t:t+k-1}$ given state $s_t$. Suppose that policy $\pi_1^k$ and $\pi_2^k$ are evaluated in the chunk-level world model $T_1^k(s'|s,\tilde{a})$ and $T_2^k(s'|s,\tilde{a})$ respectively. Assume the reward per-step satisfies $r(s_t, a_t) \in [0, r_{\max}]$, and define the $k$-steps total variation divergence between the policies as $\epsilon_\pi^k := \max_s D_{\text{TV}}\big(\pi_1^k(\tilde{a}|s) \,\|\, \pi_2^k(\tilde{a}|s)\big)$. Likewise, the dynamics divergence is bounded by $\epsilon_m^k = \max_t \mathbb{E}_{s \sim d_t^{\pi_1^k}} \big[D_{\text{TV}}\big(T_1^k(s'|s,\tilde{a})\|T_2^k(s'|s,\tilde{a})\big)\big]$. Then the value gap of two policies is bounded by*

$$\left|V(\pi_1^k) - V(\pi_2^k)\right| \leq \frac{2r_{\max}}{1-\gamma} \left[\frac{\gamma^k}{1-\gamma^k}\epsilon_\pi^k + \epsilon_\pi^k + \frac{\gamma^k}{1-\gamma^k}\epsilon_m^k\right] \tag{14}$$

*Proof.* We reformulate the problem as a *temporally-extended MDP* $\mathcal{M}^k = (\mathcal{S}, \mathcal{A}^k, \tilde{T}, \tilde{r}, \gamma^k)$, where:

- The state space $\mathcal{S}$ is unchanged.

- The action space is chunk-level $\mathcal{A}^k$, with $\tilde{a} = a_{0:k-1} \in \mathcal{A}^k$.

- The transition kernel $T^k(s' \mid s, \tilde{a})$ denotes the probability of reaching state $s'$ after executing the $k$-steps action sequence $\tilde{a}$ from state $s$.

- The reward $\tilde{r}(s, \tilde{a}) := \sum_{i=0}^{k-1} \gamma^i r(s_i, a_i)$ is the $\gamma$-discounted sum within the chunk. Since $r(s, a) \in [0, r_{\max}]$, the chunk reward is bounded by $\tilde{r}_{\max} = r_{\max}\frac{1-\gamma^k}{1-\gamma}$.

- The discount factor between macro-steps is $\gamma^k$.

Crucially, the expected return of any $k$-steps policy is identical under the original MDP and its temporally-extended variant. Let $d_t^{\pi^k}$ denote the state-action density of policy $\pi^k$ at the timestep $t$ and $d^{\pi^k} = (1-\gamma^k)\sum_{t=0}^{\infty}(\gamma^k)^t d_t^{\pi^k}$ is the overall discounted occupancy measure, such that $V(\pi^k) = \int r^k(s, \tilde{a}) \sum_{t=0}^{\infty}(\gamma^k)^t d_t^{\pi^k}(s, \tilde{a})dsd\tilde{a} = \frac{1}{1-\gamma^k}\int r^k(s, \tilde{a})d^{\pi^k}(s, \tilde{a})dsd\tilde{a}$, then we have

$$\begin{aligned}
&\left|V(\pi_1^k) - V(\pi_2^k)\right| \\
=&\frac{1}{1-\gamma^k}\left|\int r^k(s, \tilde{a})d^{\pi_1^k}(s, \tilde{a})dsd\tilde{a} - \int r^k(s, \tilde{a})d^{\pi_2^k}(s, \tilde{a})dsd\tilde{a}\right| \\
\leq&\frac{1}{1-\gamma^k}\int r^k(s, \tilde{a})\left|d^{\pi_1^k}(s, \tilde{a}) - d^{\pi_2^k}(s, \tilde{a})\right|dsd\tilde{a} \\
\leq&\frac{r_{\max}^k}{1-\gamma^k}\int \left|d^{\pi_1^k}(s, \tilde{a}) - d^{\pi_2^k}(s, \tilde{a})\right|dsd\tilde{a} \\
\leq&\frac{2r_{\max}}{1-\gamma}D_{\text{TV}}\left(d^{\pi_1^k}(s, \tilde{a})\|d^{\pi_2^k}(s, \tilde{a})\right)
\end{aligned} \tag{15}$$

By Lemma A.1, we have

$$D_{\text{TV}}(d^{\pi_1^k}(s, \tilde{a})\|d^{\pi_2^k}(s, \tilde{a})) \leq \max_s D_{\text{TV}}\left(\pi_1^k(\tilde{a}|s)\|\pi_2^k(\tilde{a}|s)\right) + D_{\text{TV}}\left(d^{\pi_1^k}(s)\|d^{\pi_2^k}(s)\right) \tag{16}$$

The first term is exactly $\epsilon_\pi^k$. For the second term, we have

$$
\begin{aligned}
D_{\text{TV}}\left(d^{\pi_1^k}(s)\|d^{\pi_2^k}(s)\right) &= \frac{1}{2}\int\left|(1-\gamma^k)\sum_{t=0}^{\infty}(\gamma^k)^t d_t^{\pi_1^k}(s) - (1-\gamma^k)\sum_{t=0}^{\infty}(\gamma^k)^t d_t^{\pi_2^k}(s)\right| ds \\
&\leq \frac{1}{2}(1-\gamma^k)\sum_{t=0}^{\infty}(\gamma^k)^t\int\left|d_t^{\pi_1^k}(s) - d_t^{\pi_2^k}(s)\right| ds \\
&= (1-\gamma^k)\sum_{t=0}^{\infty}(\gamma^k)^t D_{\text{TV}}\left(d_t^{\pi_1^k}(s)\|d_t^{\pi_2^k}(s)\right)
\end{aligned}
\tag{17}
$$

By Lemma A.2, we have

$$
\begin{aligned}
&(1-\gamma^k)\sum_{t=0}^{\infty}(\gamma^k)^t D_{\text{TV}}\left(d_t^{\pi_1^k}(s)\|d_t^{\pi_2^k}(s)\right) \\
&\leq (1-\gamma^k)\sum_{t=0}^{\infty}(\gamma^k)^t t \max_t \mathbb{E}_{s\sim d_t^{\pi_1^k}}\left[D_{\text{TV}}\left(d_t^{\pi_1^k}(s'|s)\|d_t^{\pi_2^k}(s'|s)\right)\right] \\
&\leq \frac{\gamma^k}{1-\gamma^k}\max_t \mathbb{E}_{s\sim d_t^{\pi_1^k}}\left[D_{\text{TV}}\left(d_t^{\pi_1^k}(s'|s)\|d_t^{\pi_2^k}(s'|s)\right)\right] \\
&= \frac{\gamma^k}{2(1-\gamma^k)}\max_t \mathbb{E}_{s\sim d_t^{\pi_1^k}}\left[\int\left|T_1^k(s'|s,\tilde{a})\pi_1^k(\tilde{a}|s) - T_2^k(s'|s,\tilde{a})\pi_2^k(\tilde{a}|s)\right| d\tilde{a}\right] \\
&= \frac{\gamma^k}{2(1-\gamma^k)}\max_t \mathbb{E}_{s\sim d_t^{\pi_{\mathcal{D}}^k}}\left[\int\left|T_1^k(s'|s,\tilde{a})\left(\pi_1^k(\tilde{a}|s) - \pi_2^k(\tilde{a}|s)\right) + \pi_2^k(\tilde{a}|s)\left(T_1^k(s'|s,\tilde{a}) - T_2^k(s'|s,\tilde{a})\right)\right| d\tilde{a}\right] \\
&\leq \frac{\gamma^k}{2(1-\gamma^k)}\max_t \mathbb{E}_{s\sim d_t^{\pi_{\mathcal{D}}^k}}\left[\int\left|\pi_1^k(\tilde{a}|s) - \pi_2^k(\tilde{a}|s)\right| d\tilde{a} + \int\left|T_1^k(s'|s,\tilde{a}) - T_2^k(s'|s,\tilde{a})\right| d\tilde{a}\right] \\
&\leq \frac{\gamma^k}{(1-\gamma^k)}\left(\max_s D_{\text{TV}}\left(\pi_1^k(\tilde{a}|s)\|\pi_2^k(\tilde{a}|s)\right) + \max_t \mathbb{E}_{s\sim d_t^{\pi_1^k}}\left[D_{\text{TV}}\left(T_1^k(s'|s,\tilde{a})\|T_2^k(s'|s,\tilde{a})\right)\right]\right) \\
&\leq \frac{\gamma^k(\epsilon_\pi^k + \epsilon_m^k)}{(1-\gamma^k)}
\end{aligned}
\tag{18}
$$

Finally, we have

$$
\left|V(\pi_1^k) - V(\pi_2^k)\right| \leq \frac{2r_{\max}}{1-\gamma}D_{\text{TV}}(d^{\pi_1^k}\|d^{\pi_2^k}) \leq \frac{2r_{\max}}{1-\gamma}\left(\epsilon_\pi^k + \frac{\gamma^k(\epsilon_\pi^k + \epsilon_m^k)}{(1-\gamma^k)}\right) = \frac{2\gamma^k r_{\max}(\epsilon_\pi^k + \epsilon_m^k)}{(1-\gamma)(1-\gamma^k)} + \frac{2r_{\max}\epsilon_\pi^k}{1-\gamma}
\tag{19}
$$

$\square$

**Lemma A.4** (Value Divergence, Chunk-level Policy, Chunk-level World Model, Branched Rollout). *Let $k \in \mathbb{N}^+$ be the output chunk size of policies. Consider two stochastic $k$-step policies $\pi_1^k$ and $\pi_2^k$, where each policy selects an action sequence $\tilde{a}_t = a_{t:t+k-1}$ given state $s_t$. Suppose that policy $\pi_1^k$ and $\pi_2^k$ are evaluated in the chunk-level world model $T_1^k(s'|s,\tilde{a})$ and $T_2^k(s'|s,\tilde{a})$ respectively. Assume we run a branched rollout of length $n$ with chunk size $k$. Before the branch ("pre" branch), we assume that the expected total variation divergence of dynamics is bounded as $\max_t \mathbb{E}_{s\sim d_t^{\pi_1^k}} D_{\text{TV}}\left(T_1^{k,pre}(s'|s,\tilde{a})\|T_2^{k,pre}(s'|s,\tilde{a})\right) \leq \epsilon_m^{k,pre}$ and after the branch as $\max_t \mathbb{E}_{s\sim d_t^{\pi_1^k}} D_{\text{TV}}\left(T_1^{k,post}(s'|s,\tilde{a})\|T_2^{k,post}(s'|s,\tilde{a})\right) \leq \epsilon_m^{post}$. Likewise, the policy divergence is bounded pre-branch and post-branch by $\epsilon_\pi^{k,pre}$ and $\epsilon_\pi^{k,post}$, respectively. Then, the value gap of policy $\pi_1^k$ and $\pi_2^k$ is bounded as:*

$$
\left|V(\pi_1^k) - V(\pi_2^k)\right| \leq \frac{2r_{\max}}{1-\gamma}\left[n\epsilon_m^{k,post} + (n+1)\epsilon_\pi^{k,post} + \frac{(\gamma^k)^{n+1}(\epsilon_m^{k,pre} + \epsilon_\pi^{k,pre})}{1-\gamma^k} + (\gamma^k)^n\epsilon_\pi^{k,pre}\right]
\tag{20}
$$

*Proof.* We again utilize the temporally-extended MDP $\mathcal{M}^k = (\mathcal{S}, \mathcal{A}^k, \tilde{T}, \tilde{r}, \gamma^k)$. In this view, a branched rollout of length $n$ chunks starting from $s \sim d^{\pi_1^k}$ corresponds to a standard trajectory in $\mathcal{M}^k$ with discount factor $\gamma^k$, but truncated or branched.

Similar to Equation 15 and Equation 17, we obtain

$$\left|V(\pi_1^k) - V(\pi_2^k)\right| \leq \frac{2r_{\max}}{1-\gamma} D_{\mathrm{TV}}\left(d^{\pi_1^k}(s,\tilde{a})\|d^{\pi_2^k}(s,\tilde{a})\right) \leq \frac{2r_{\max}}{1-\gamma}(1-\gamma^k)\sum_{t=0}^{\infty}(\gamma^k)^t D_{\mathrm{TV}}\left(d_t^{\pi_1^k}(s,\tilde{a})\|d_t^{\pi_2^k}(s,\tilde{a})\right) \quad (21)$$

For $t \leq n$, by Lemma A.1 and Lemma A.2, we have

$$D_{\mathrm{TV}}\left(d_t^{\pi_1^k}(s,\tilde{a})\|d_t^{\pi_2^k}(s,\tilde{a})\right) \leq D_{\mathrm{TV}}\left(d_t^{\pi_1^k}(s)\|d_t^{\pi_2^k}(s)\right) + \max_s D_{\mathrm{TV}}\left(\pi_1^k(\tilde{a}|s)\|\pi_2^k(\tilde{a}|s)\right) \leq t(\epsilon_m^{k,\mathrm{post}} + \epsilon_\pi^{k,\mathrm{post}}) + \epsilon_\pi^{k,\mathrm{post}} \quad (22)$$

And for $t \geq n$ we have

$$D_{\mathrm{TV}}\left(d_t^{\pi_1^k}(s,\tilde{a})\|d_t^{\pi_2^k}(s,\tilde{a})\right) \leq n(\epsilon_m^{k,\mathrm{post}} + \epsilon_\pi^{k,\mathrm{post}}) + (t-n)(\epsilon_m^{k,\mathrm{pre}} + \epsilon_\pi^{k,\mathrm{pre}}) + \epsilon_\pi^{k,\mathrm{post}} + \epsilon_\pi^{k,\mathrm{pre}} \quad (23)$$

Combined the results above, we have

$$\begin{aligned}
&\frac{2r_{\max}(1-\gamma^k)}{1-\gamma}\sum_{t=0}^{\infty}(\gamma^k)^t\left(D_{\mathrm{TV}}\left(d_t^{\pi_1^k}(s,\tilde{a})\|d_t^{\pi_2^k}(s,\tilde{a})\right)\right) \\
\leq & \frac{2r_{\max}(1-\gamma^k)}{1-\gamma}\left(\sum_{t=0}^{n}(\gamma^k)^t D_{\mathrm{TV}}\left(d_t^{\pi_1^k}(s,\tilde{a})\|d_t^{\pi_2^k}(s,\tilde{a})\right) + \sum_{t=n+1}^{\infty}(\gamma^k)^t D_{\mathrm{TV}}\left(d_t^{\pi_1^k}(s,\tilde{a})\|d_t^{\pi_2^k}(s,\tilde{a})\right)\right) \\
\leq & \frac{2r_{\max}(1-\gamma^k)}{1-\gamma}\sum_{t=0}^{n}(\gamma^k)^t\left(t(\epsilon_m^{k,\mathrm{post}} + \epsilon_\pi^{k,\mathrm{post}}) + \epsilon_\pi^{k,\mathrm{post}}\right) \\
& + \frac{2r_{\max}(1-\gamma^k)}{1-\gamma}\sum_{t=n+1}^{\infty}(\gamma^k)^t\left(n(\epsilon_m^{k,\mathrm{post}} + \epsilon_\pi^{k,\mathrm{post}}) + (t-n)(\epsilon_m^{k,\mathrm{pre}} + \epsilon_\pi^{k,\mathrm{pre}}) + \epsilon_\pi^{k,\mathrm{post}} + \epsilon_\pi^{k,\mathrm{pre}}\right) \\
\leq & \frac{2r_{\max}(1-\gamma^k)}{1-\gamma}\sum_{t=0}^{\infty}(\gamma^k)^t\left(n(\epsilon_m^{k,\mathrm{post}} + \epsilon_\pi^{k,\mathrm{post}}) + \epsilon_\pi^{k,\mathrm{post}}\right) \\
& + \frac{2r_{\max}(1-\gamma^k)}{1-\gamma}\sum_{t=n+1}^{\infty}(\gamma^k)^t\left((t-n)(\epsilon_m^{k,\mathrm{pre}} + \epsilon_\pi^{k,\mathrm{pre}}) + \epsilon_\pi^{k,\mathrm{pre}}\right) \\
\leq & \frac{2r_{\max}(\epsilon_m^{k,\mathrm{post}} + \epsilon_\pi^{k,\mathrm{post}})}{1-\gamma} + \frac{2r_{\max}\epsilon_\pi^{k,\mathrm{post}}}{1-\gamma} + \frac{2r_{\max}(\gamma^k)^{n+1}(\epsilon_m^{k,\mathrm{pre}} + \epsilon_\pi^{k,\mathrm{pre}})}{(1-\gamma)(1-\gamma^k)} + \frac{2r_{\max}(\gamma^k)^n\epsilon_\pi^{k,\mathrm{pre}}}{1-\gamma}
\end{aligned} \quad (24)$$

Simplify the inequality above, finally we have

$$\left|V(\pi_1^k) - V(\pi_2^k)\right| \leq \frac{2r_{\max}}{1-\gamma}\left[n\epsilon_m^{k,\mathrm{post}} + (n+1)\epsilon_\pi^{k,\mathrm{post}} + \frac{(\gamma^k)^{n+1}(\epsilon_m^{k,\mathrm{pre}} + \epsilon_\pi^{k,\mathrm{pre}})}{1-\gamma^k} + (\gamma^k)^n\epsilon_\pi^{k,\mathrm{pre}}\right] \quad (25)$$

$\square$

## B. Theoretical Analysis

**Theorem B.1** (Value Gap, Chunk-Level Policy, Step-Level World Model). *Given an offline dataset $\mathcal{D}$ and a chunk size $k \in \mathbb{N}^+$, where the policy selects an action sequence $\tilde{a}_t = a_{t:t+k-1}$ given state $s_t$. Assume the reward per-step satisfies $r(s_t, a_t) \in [0, r_{\max}]$, and define the $k$-step total variation divergence between the policy $\pi^k$ and the behavior policy $\pi_{\mathcal{D}}^k$ is bounded by $\epsilon_\pi^k = \max_s D_{\mathrm{TV}}\left(\pi_{\mathcal{D}}^k(\tilde{a}|s)\|\pi^k(\tilde{a}|s)\right)$. Let $\epsilon_m = \max_t \mathbb{E}_{s \sim \mathcal{D}^t}\left[D_{\mathrm{TV}}\left(T(s'|s,a)\|\hat{T}(s'|s,a)\right)\right]$ denote the expected total variation divergence of learned dynamics $\hat{T}$. Then, the value gap of policy $\pi^k$ under the true environment and the learned world model is bounded by:*

$$\left|V(\pi^k) - \hat{V}(\pi^k)\right| \leq \frac{2r_{\max}}{1-\gamma}\left[\frac{2\gamma^k\epsilon_\pi^k}{1-\gamma^k} + 2\epsilon_\pi^k + \frac{k\gamma^k}{1-\gamma^k}\epsilon_m\right] \quad (26)$$

*Proof.* We decompose the value gap as follows:

$$
\begin{aligned}
&|V(\pi^k) - \hat{V}(\pi^k)| \\
=&|V(\pi^k) - V(\pi^k_{\mathcal{D}}) + V(\pi^k_{\mathcal{D}}) - \hat{V}(\pi^k)| \\
\leq&\underbrace{|V(\pi^k) - V(\pi^k_{\mathcal{D}})|}_{L_1} + \underbrace{|V(\pi^k_{\mathcal{D}}) - \hat{V}(\pi^k)|}_{L_2}
\end{aligned}
\tag{27}
$$

For $L_1$, we apply Lemma A.3 with zero dynamics divergence and obtain

$$
L_1 \leq \frac{2r_{\max}}{1-\gamma}\left[\frac{\gamma^k}{1-\gamma^k}\epsilon^k_\pi + \epsilon^k_\pi\right]
\tag{28}
$$

For $L_2$, we notice that the chunk-level divergence $\epsilon^k_m = \max_t \mathbb{E}_{s \sim d^{\pi^k}_t}\left[D_{\mathrm{TV}}\left(T^k(s'|s,\tilde{a})\|\hat{T}^k(s'|s,\tilde{a})\right)\right]$ of step-level world models $T$ and $\hat{T}$ is bounded by $k\epsilon_m$, then we apply Lemma A.3 again and obtain

$$
L_2 \leq \frac{2r_{\max}}{1-\gamma}\left[\frac{\gamma^k}{1-\gamma^k}\epsilon^k_\pi + \epsilon^k_\pi + \frac{k\gamma^k}{1-\gamma^k}\epsilon_m\right]
\tag{29}
$$

Finally, combined with the $L_1$ and $L_2$ we have

$$
|V(\pi^k) - \hat{V}(\pi^k)| \leq \frac{2r_{\max}}{1-\gamma}\left[\frac{2\gamma^k\epsilon^k_\pi}{1-\gamma^k} + 2\epsilon^k_\pi + \frac{k\gamma^k}{1-\gamma^k}\epsilon_m\right]
\tag{30}
$$

$\square$

**Theorem B.2** (Value Gap, Chunk-level Policy, Chunk-level World Model, Branched Rollout). *Given an offline dataset $\mathcal{D}$ and chunk size $k \in \mathbb{N}^+$, where the policy selects an action sequence $\tilde{a} = a_{t:t+k-1}$ given state $s_t$. Suppose that policy $\pi^k$ is evaluated via $n$-chunks branched rollout in the chunk-level world model $\hat{T}^k(s'|s,\tilde{a})$. Assume the reward per-step satisfies $r(s_t, a_t) \in [0, r_{\max}]$, and define the $k$-steps total variation divergence between the policies as $\epsilon^k_\pi = \max_s D_{\mathrm{TV}}\left(\pi^k_{\mathcal{D}}(\tilde{a}|s) \,\|\, \pi^k(\tilde{a}|s)\right)$. Moreover, assume that the expected total variation divergence of dynamics **on the current policy branched rollout** is defined as $\epsilon^{k,n}_m = \max_{t \leq n} \mathbb{E}_{s \sim d^{\pi^k}_{t,s_0 \sim \mathcal{D}}}\left[D_{\mathrm{TV}}\left(T^k(s'|s,\tilde{a})\|\hat{T}^k(s'|s,\tilde{a})\right)\right]$. Then the value gap of policy $\pi^k$ under the true environment and the learned world model is bounded by:*

$$
\left|V(\pi^k) - \hat{V}^{branch}(\pi^k)\right| \leq \frac{2r_{\max}}{1-\gamma}\left[\frac{(\gamma^k)^{n+1}}{1-\gamma^k}\epsilon^k_\pi + (\gamma^k)^n\epsilon^k_\pi + n\epsilon^{k,n}_m\right]
\tag{31}
$$

*Proof.* We define $\hat{V}(\pi^k_{\mathcal{D}}, \pi^k)^{\mathrm{branch}}$ as the value of which executes the $\pi^k_{\mathcal{D}}$ under the true dynamics until the branch point, then executes $\pi^k$ under the true dynamics for $n$-chunks. Then we decompose the value gap as follows:

$$
\left|V(\pi^k) - \hat{V}^{\mathrm{branch}}(\pi^k)\right| \leq \underbrace{\left|V(\pi^k) - \hat{V}^{\mathrm{branch}}(\pi^k_{\mathcal{D}}, \pi^k)\right|}_{L_1} + \underbrace{\left|\hat{V}^{\mathrm{branch}}(\pi^k_{\mathcal{D}}, \pi^k) - V^{\mathrm{branch}}(\pi^k)\right|}_{L_2}
\tag{32}
$$

$\square$

For $L_1$, we notice that this term only suffers from error before the branch begins, and we use Lemma B.2 with $\epsilon^{k,\mathrm{pre}}_\pi \leq \epsilon^k_\pi$ and set all other errors to 0, then we have

$$
L_1 \leq \frac{2r_{\max}}{1-\gamma}\left[\frac{(\gamma^k)^{n+1}}{1-\gamma^k}\epsilon^k_\pi + (\gamma^k)^n\epsilon^k_\pi\right]
\tag{33}
$$

For $L_2$, it incorporates model error under the policy $\pi^k$ incurred after the branch. Again we use Lemma B.2 with $\epsilon^k_{m'}$ set all other errors to 0, we obtain

$$
L_2 \leq \frac{2r_{\max}}{1-\gamma}n\epsilon^{k,n}_m
\tag{34}
$$

Adding $L_1$ and $L_2$, we have

$$\left| V(\pi^k) - \hat{V}^{\text{branch}}(\pi^k) \right| \leq \frac{2r_{\max}}{1-\gamma} \left[ \frac{(\gamma^k)^{n+1}}{1-\gamma^k} \epsilon_\pi^k + (\gamma^k)^n \epsilon_\pi^k + n\epsilon_m^{k,n} \right] \tag{35}$$

## C. Real-World Experiments

### C.1. Hardware Setup

We evaluate our method on two distinct real-world robotic platforms: the Arx-X5 bimanual robot (left) and the Galaxy-R1 whole-body robot (right) as shown in figure 7.

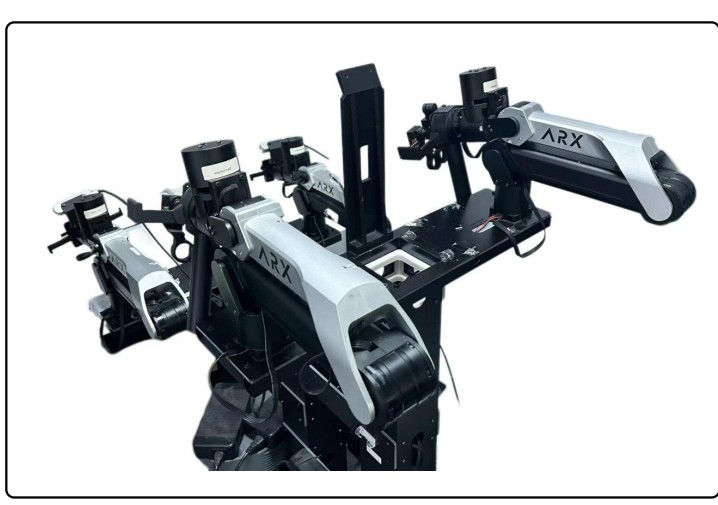 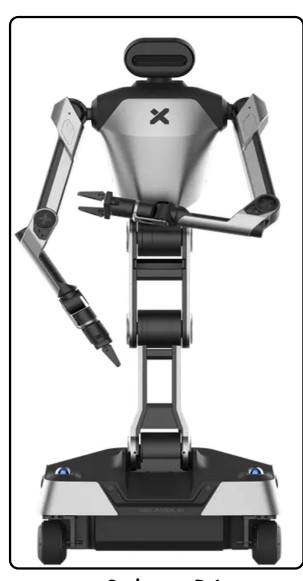

ARX-X5                                   Galaxy R1

*Figure 7.* An illustration of real-world robots.

**Galaxy-R1 System**   The Galaxy-R1 is a high-dimensional whole-body robot featuring a 21-DoF kinematic structure, comprising dual 7-DoF arms, a 4-DoF torso, and a 3-DoF mobile base. It is equipped with a ZED 2i stereo camera (head) and two Intel RealSense D435i cameras (wrists), utilizing Galaxea G1 parallel grippers for manipulation. Data collection is performed via the JoyLo teleoperation system, which integrates 3D-printable links and Dynamixel actuators with Joy-Con controllers; the control loop operates at 100 Hz, with data synchronized and recorded at 10 Hz.

**Arx-X5 System**   The Arx-X5 is a bimanual platform with a dual-arm configuration totaling 14 DoF. Its perception suite consists of three Intel RealSense D435i RGB-D cameras providing multi-view feedback. For data collection, we employ a master-slave teleoperation setup, where two handheld master arms directly control the robot's follower arms. The system operates at a control frequency of 15 Hz, which is sufficient for the quasi-static manipulation tasks utilized in our experiments.

**Real-world Assets.**   We present the comprehensive set of physical assets employed in our experiments in Figure 8. The left panel displays the seen objects that were accessible to the robot during the training phase, while the right panel introduces the unseen objects.

**Task Illustration.**   As illustrated in Figure 9, our evaluation suite encompasses five distinct tasks designed to test precision, deformable object manipulation, and whole-body coordination: (a) **Plug In**: A high-precision task requiring the robot to align and insert a plug into a power strip socket. (b) **Fold Towel**: A long-horizon deformable object task. Conditioned on a language prompt, the robot selects the correct towel from two options, drags it to the workspace center, performs an initial bimanual fold by grasping the top corners, executes a second fold by coordinating a left-hand hold with a right-hand grasp,

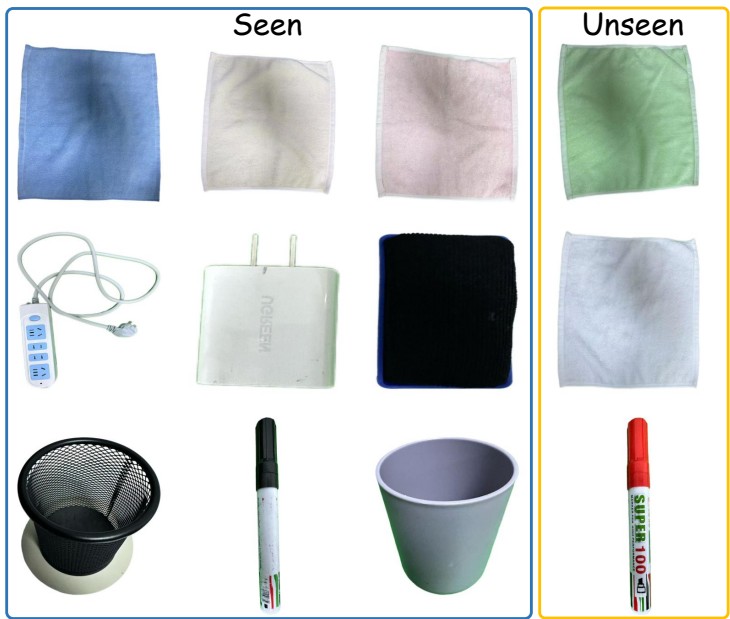

*Figure 8.* An illustration of real world assets.

and finally positions the folded towel. (c) **Insert Pen**: Requires fine motor skills to pick up a pen from the table and insert it into a narrow pen holder. (d) **Pick Cup**: A pick-and-place task where the robot grasps a cup and precisely deposits it onto a target plate. (e) **Wipe Board**: A whole-body mobile manipulation task requiring the robot to execute a 90° base rotation, navigate towards a whiteboard, and raise its end-effector to erase specific visual targets.

**Data Collection and Policy Evaluation**    For all tasks, our data collection strategy involves an initial phase of human-teleoperated expert demonstrations for Supervised Fine-Tuning (SFT), followed by a mix of expert and autonomous self-collected data for Reinforcement Learning (RL). Evaluation of each tasks comprises 50 trials divided into 30 seen and 20 unseen socket configurations The specific protocols for each task are as follows:

- **Plug In:** The SFT dataset consists of 50 expert trajectories. For RL, we augment this with 50 self-collected on-policy trajectories. The test scenarios are specifically designed to probe the policy's spatial generalization capabilities in high-precision alignment scenarios.

- **Fold Towel:** We collect 60 expert trajectories for SFT, covering 3 distinct towel colors (20 trajectories each). The RL dataset combines these expert data with 50 self-collected trajectories. The evaluation protocol tests robustness against background shifts, instruction variations, and novel towel instances, assessing the model's fundamental understanding of deformable object dynamics.

- **Insert Pen:** The SFT phase utilizes 100 expert trajectories, while the RL phase adds 50 self-collected trajectories. We conduct evaluation focusing on high-dimensional spatial understanding, specifically examining the policy's ability to handle object generalization and severe viewpoint disturbances during fine manipulation.

- **Pick Cup:** Similar to the pen task, we use 100 expert trajectories for SFT and add 50 self-collected trajectories for RL. The 50 evaluation trials are designed to test visual generalization under active camera motion, specifically assessing stability when the torso and mobile base introduce significant viewpoint perturbations.

- **Wipe Board:** We employ 100 expert trajectories for SFT and supplement this with 50 self-collected trajectories for RL. The evaluation targets robustness under partial observability and high compounding errors arising from navigation. It further tests generalization across stain variations, including color, position, and quantity.

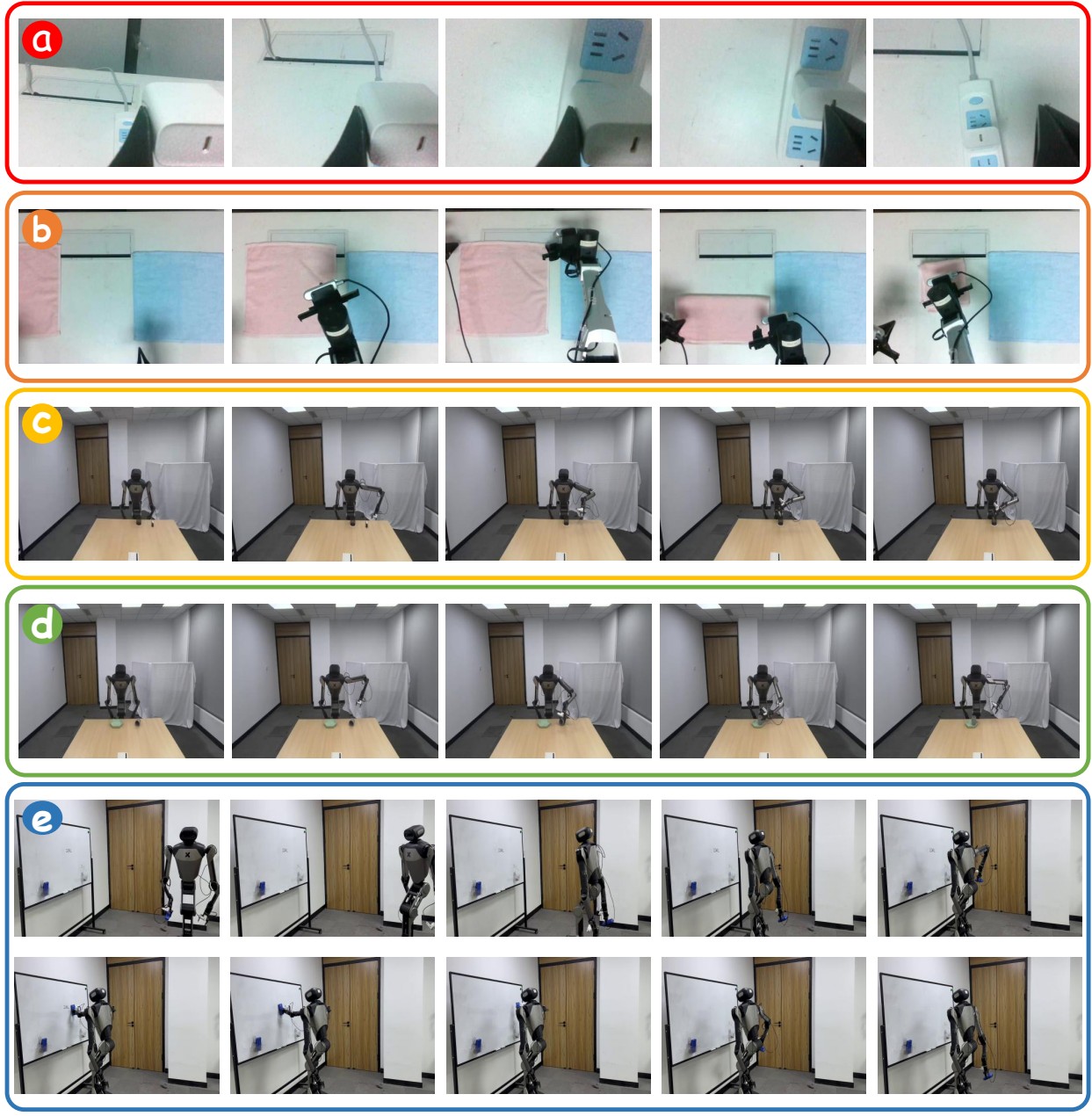

*Figure 9.* An illustration of real world tasks. We test VLA-MBPO on bimanual and whole-body robots.

# D. Implementation Details

**Causal Attention Mask Implementation**   Figure 10 illustrates the design of our unified causal attention mask, which governs the information flow between multi-modal tokens during training. The sequence integrates visual observations, action sequences and text instructions. The token sequence consists of ViT and VAE features. ViT represents high-level semantic features extracted by the vision encoder to provide semantic grounding, while VAE denotes the compressed latent codes used for high-fidelity image reconstruction. The sequence is further structured by control tokens such as "Action Chunk" (representing the robot's discrete control signals) and task-specific prompts. We present

**Prompt Engineering**   To align the model's generation with specific task requirements, we utilize structured text prompts for both world model and reward model prediction. The exact prompt templates are detailed below.

- **World Model Prompt.** We instruct the model to act as a physical simulator. The input integrates the history of observations with a discrete action sequence, where the action sequence is normalized into integer arrays representing joint positions and gripper states for each timestep (e.g., `[111, 113, ...]`).

   *"You are now acting as a **world model** that simulates robot manipulation task execution. Your task is to predict the **next frame of visual observation**, given the following inputs: (1) Multiple current observation images from the robot's cameras (head and wrist); (2) An action sequence describing the manipulation to execute; (3) Optionally, the next frame from the head camera (for predicting wrist camera views). You will receive images from different camera viewpoints and need to predict the next frame according to the provided action sequence and instruction."*

   Following this system instruction, the model receives the action chunk and the specific generation command, for example: `"...  Step 9:  [124, 34, 127, 133, 241, 129, 0].  Predict next head camera view according to the current observation and action."`

- **Reward Model Prompt.** For the reward model, we leverage a pre-trained VLM as a success detector. We enforce a strict binary output format (Yes or No).

   *"You are a vision-language model with advanced reasoning abilities. Your task is to carefully observe the image and determine whether the task is successfully completed.*
   ***Environment description:** You are observing a robot workspace from the LIBERO dataset. The robot can manipulate objects (pick, place, arrange) in the scene.*
   ***Guidelines:** Carefully examine the state of objects. Check if the goal state matches the task description. Consider spatial arrangement.*
   ***Response format:** Answer with "Yes." if the task is successfully completed. Answer with "No." if the task is not yet completed or failed. Your response must be either "Yes." or "No." without additional explanation.*
   ***Task:** Determine whether the task: [Task Description] is successfully completed."*

**Hyperparameters.**   We present the detailed training hyperparameters for the UMM-World and the downstream policy optimization in Table 4 and Table 5. Respectively. A key advantage of our framework is its stability and ease of tuning; as evidenced in Table 5, the majority of hyperparameters remain constant across all task suites. The primary adaptation required is scaling the Sample Size (and the corresponding Update to data steps) for long horizon tasks.

*Table 4.* Hyperparameters for UMM-World.

| Parameters | Value |
|---|---|
| lr | 2e−5 |
| Cross entropy weight | 0.01 |
| MSE weight | 1.0 |
| Max latent size | 64 |
| Timestep shift | 4 |
| Cfg interval | [0.4, 1.0] |
| Text scale | 6 |
| Image scale | 2 |

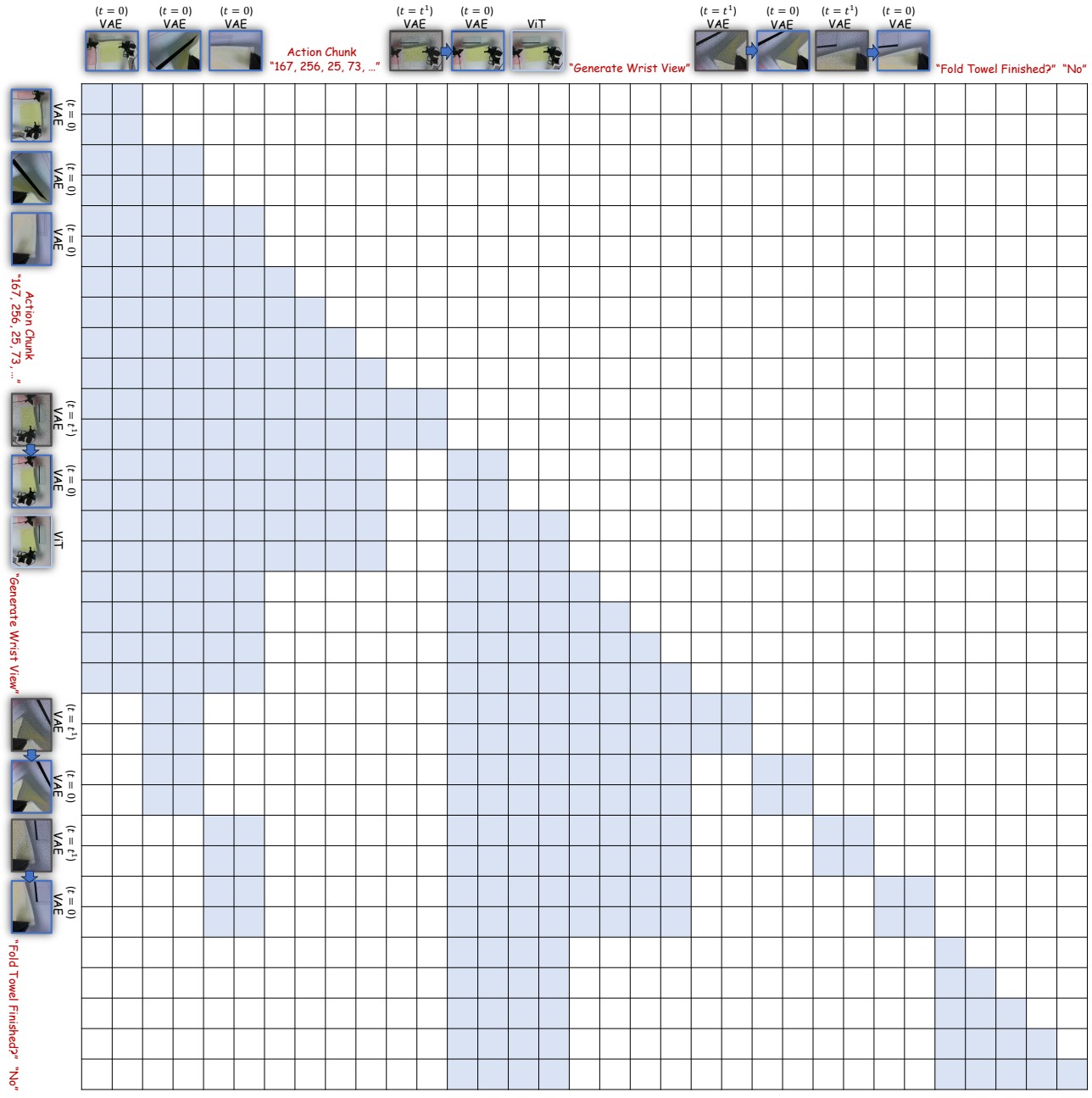

*Figure 10.* **Causal Mask in UMM World Model during training**. VAE and ViT denote VAE features and ViT features, respectively. $t$ is the noise timestep and $t = 0$ means no noise.

*Table 5.* Hyperparameters for RL.

| Parameters | Libero | | | | Real-World |
|---|---|---|---|---|---|
| | **Spatial** | **Object** | **Goal** | **Long** | |
| Sample size | 512 | 512 | 512 | 1280 | 512 |
| Batch size | 512 | 512 | 512 | 512 | 512 |
| Rollout chunk | 2 | 2 | 2 | 2 | 2 |
| Actor lr | 5e−6 | 5e−6 | 5e−6 | 5e−6 | 5e−6 |
| Critic lr | 1e−4 | 1e−4 | 1e−4 | 1e−4 | 1e−4 |
| Reward discount rate $\gamma$ | 0.99 | 0.99 | 0.99 | 0.99 | 0.99 |
| GAE $\lambda$ | 0.95 | 0.95 | 0.95 | 0.95 | 0.95 |
| Clip ratio $\epsilon$ | 0.1 | 0.1 | 0.1 | 0.1 | 0.1 |
| Action chunk $H$ | 10 | 10 | 10 | 10 | 10 |
| Denoise steps | 3 | 3 | 3 | 3 | 3 |
| Noise level | 0.5 | 0.5 | 0.5 | 0.5 | 0.5 |
| Update to data | 20 | 20 | 20 | 50 | 20 |

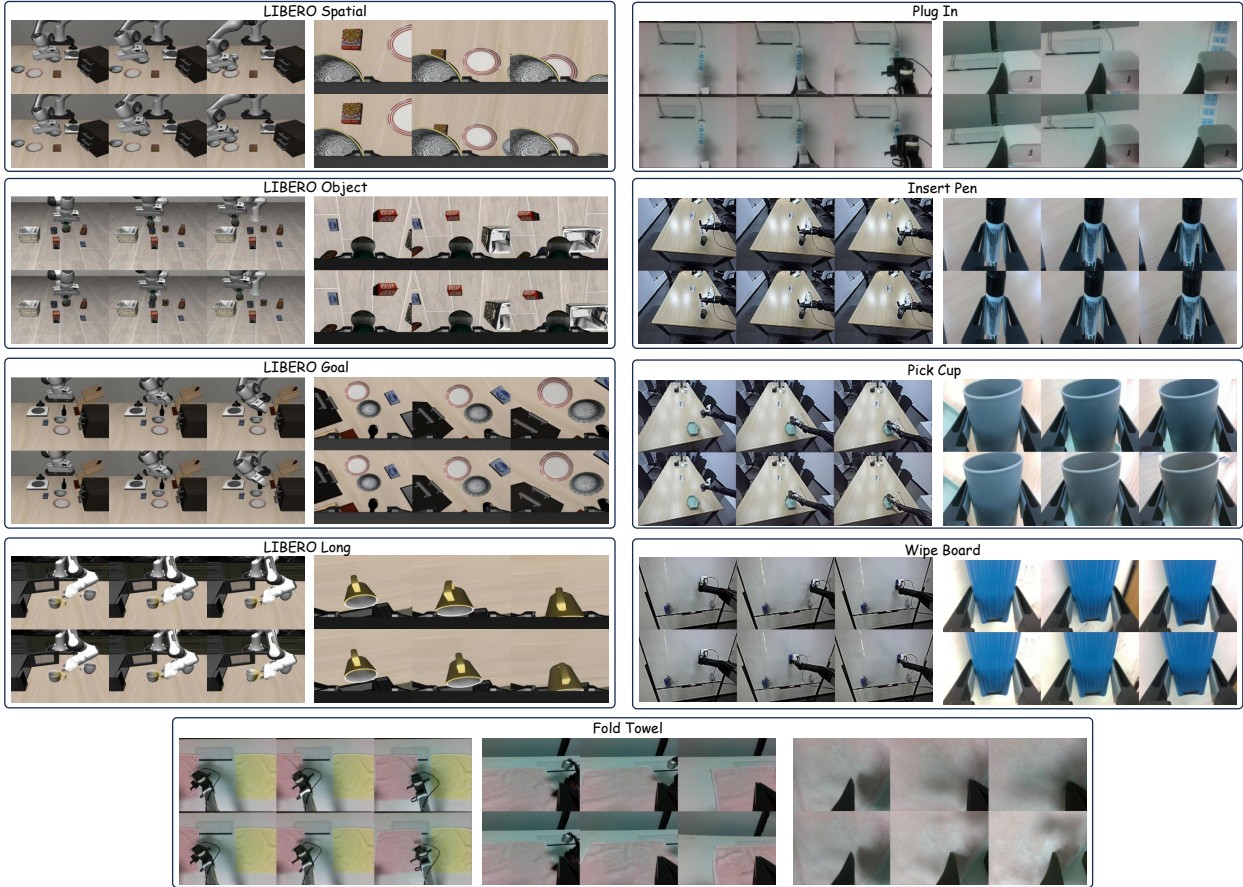

*Figure 11.* Qualitative comparison of UMM-World prediction against Ground Truth across both simulated (LIBERO) and real-world tasks.

# E. Additional Results

### E.1. World Model Generation Visualization

**Visualization Analysis** Figure 11 provides a qualitative comparison between the generated trajectories of UMM-World and the ground truth. The visualization is organized with the initial observation in the leftmost column, followed by the predicted sequence spanning 20 steps for simulated tasks and 40 steps for real-world scenarios.

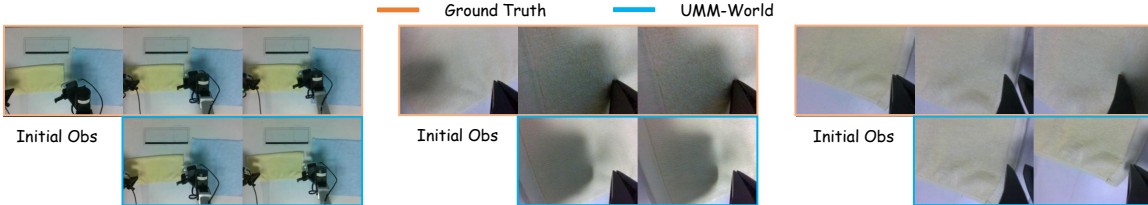

*Figure 12.* An illustration showing spatial consistency of UMM-World generation.

**Case Study** Figure 12 illustrates UMM-World's internal coherence through two distinct qualitative examples spanning two consecutive 20-step chunks. In the left deformable object included task, the model exhibits strong geometric consistency. Although the predicted head view diverges from the ground truth regarding the right hand's position, the generated wrist view aligns with the predicted head view rather than the ground truth. This confirms that our interleaved decoding strategy enforces strict spatial coherence, ensuring the body posture remains consistent across viewpoints. The right example highlights the model's adherence to physical laws. While the ground truth shows the robot erasing the entire word in the first chunk, the model predicts a variation where the letter E' remains un-erased. Crucially, both the generated head and wrist views consistently depict this incomplete' state. This demonstrates that the model maintains a unified physical reality—correctly reflecting the remaining ink across all sensors—rather than simply memorizing the expert's outcome.

### E.2. Failure Case Analysis

1. **Partially Observable.** As illustrated in Figure 13, we analyze UMM-World's failure cases from inherent partial observability in embodied control. The left panel demonstrates a failure during the arm lifting phase in the *Wipe Board* task. As the manipulator moves out of the head camera's field of view, the model fails to render the arm in the predicted head view. Although the wrist camera provides local visual feedback, the model struggles to infer the global arm posture solely from these local cues without explicit kinematic history in the main view, leading to a disappearance of the robot arm in the generated future.The right panel depicts a failure during the navigation phase involving significant viewpoint changes. When the robot rotates its base to face the whiteboard, the target object enters the field of view for the first time. Since this region was never present in the condition observations, the world model cannot hallucinate the correct geometric structure and texture of the unseen environment from nothing.

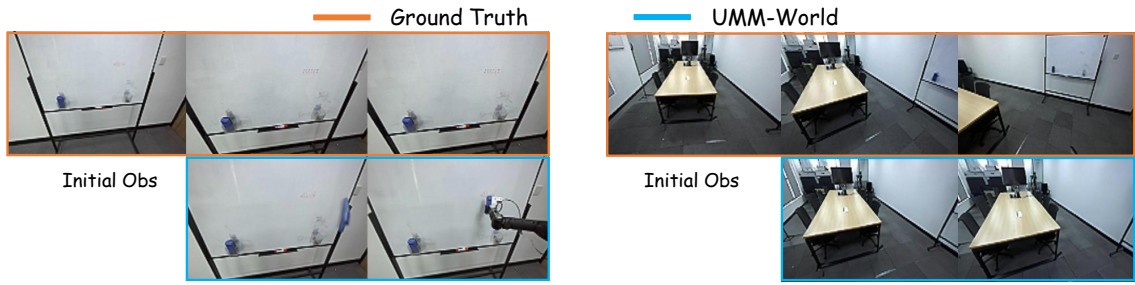

*Figure 13.* An illustration of failure case caused of partial observability.

2. **Large Physical Movements.** As shown in Figure 14, we observe generation failures when the target sequence involves excessive physical movement or state changes. The left panel illustrates a *Wipe Board* sequence characterized by significant end-effector displacement. Due to the difficulty of modeling such large-scale motion in a single shot, the

model suffers from motion collapse; instead of capturing the extensive trajectory, it conservatively predicts minimal change, resulting in a generated arm that appears nearly static despite the actual dynamic motion. Similarly, in the *Fold Towel* task (right panel), the sequence encompasses drastic configuration changes inherent to interacting with deformable objects. The complexity and stochasticity of these large-scale non-rigid transitions prove difficult to bridge accurately. Consequently, the model succumbs to hallucination, generating physically implausible towel configurations rather than correctly tracking the substantial deformation process.

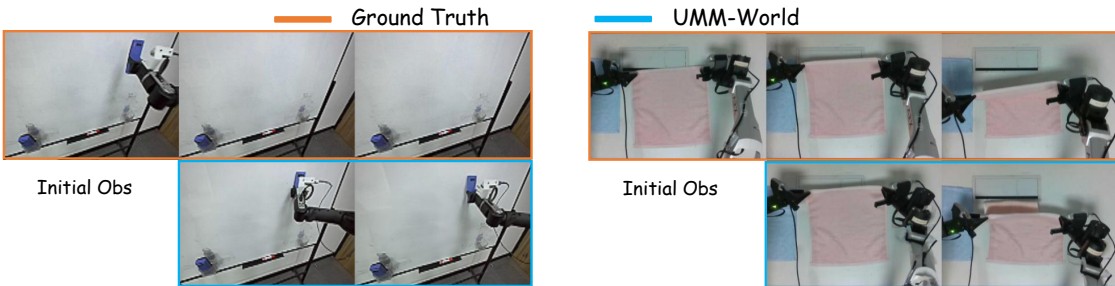

*Figure 14.* An illustration of failure cases caused by large prediction horizon.

## F. Computational Resources

All experiments were conducted on 8 NVIDIA H100 GPUs. The computational cost for UMM-World training on the collected dataset requires approximately 7-8 hours. The computational cost for policy optimization typically completes in approximately 4-6 hours under the standard interaction budget.

