# OpenReview forum: "Towards Practical World Model-based Reinforcement Learning for Vision-Language-Action Models"
_ICML.cc/2026/Conference — ICML 2026 regular_

### Official Review · Reviewer_RgpN · 2026-03-08

**Soundness:** 3
**Presentation:** 3
**Significance:** 3
**Originality:** 2
**Overall Recommendation:** 5
**Confidence:** 3

**Summary:**

This paper proposes VLA-MBPO, a world model-based reinforcement learning framework for finetuning Vision-Language-Action models without real-world interaction. To address the challenges of pixel-level world modeling, multi-view consistency, and compounding errors under sparse rewards, VLA-MBPO integrates a Unified Multimodal Model (UMM) as the world model backbone, an interleaved view decoding strategy for cross-view consistency, and chunk-level branched rollouts to mitigate error accumulation. The framework is validated on the LIBERO benchmark and 5 real-world tasks across two robotic platforms, accompanied by theoretical analysis of value gap reduction.

**Compliance With Llm Reviewing Policy:**

Affirmed.

**Final Justification:**

My main concerns in the initial review were: (1) limited novelty beyond systems integration, (2) lack of OOD evaluation for action-conditioned predictions, and (3) missing error bars. The rebuttal adequately addressed (2) and (3) by providing OOD generation results on RoboTwin, standard deviations across seeds, a unified world model scaling experiment, and real-world reward model evaluation. Regarding (1), I accept the authors' argument that a principled, minimal framework that works robustly without task-specific tuning is a meaningful practical contribution, though the originality remains fair.
The remaining gap is that both the OOD and unified world model evaluations only report generation-level metrics without downstream policy performance, which I encourage the authors to include in the camera-ready. Overall, I believe the paper makes a solid contribution to VLA finetuning and raise my recommendation from 4 to 5: Accept.

**Key Questions For Authors:**

1. The paper does not clearly state whether the world model is trained per task or per suite. From my understanding of the code, it appears that a separate world model is finetuned for each task suite. Have you explored training a single world model across multiple suites or domains? This would strengthen the scalability claims and is a natural next step given the UMM backbone's pretraining on diverse data.

2. The framework relies on an offline dataset for both world model finetuning and branched rollout starting states. How does performance degrade if the offline trajectories are suboptimal or cover only a narrow portion of the state space?

3. Table 1 evaluates reward prediction only on LIBERO. How accurate is the UMM-based reward model on the real-world tasks, where visual complexity and domain shift from pretraining data are greater?

**Limitations:**

yes

**Strengths And Weaknesses:**

**Strengths**

1. The three proposed components form a coherent system design and yield notable performance gains. The UMM's chunk-level prediction aligns with action chunking in VLA policies, enabling efficient branched rollouts that reduce both inference cost and error accumulation. The interleaved view decoding leverages the autoregressive structure of the UMM. The design choices are well-motivated and clearly presented.

2. VLA-MBPO maintains nearly identical hyperparameters across all four LIBERO suites and all real-world tasks. This largely reduces the engineering effort required when applying the framework to new tasks, which is an important consideration for real-world deployment.

3. The experimental coverage is thorough. The experiments span both simulation (LIBERO, 4 suites) and real-world tasks (2 robotic platforms, 5 tasks), covering a diverse range of manipulation skills. The world model is also separately evaluated on dynamics prediction, reward prediction, and inference speed.

**Weaknesses**

1. Finetuning a pretrained multimodal model into an action-conditioned world model is a straightforward idea, and the branched rollout strategy largely follows MAC [1]. The theoretical analysis is a relatively straightforward extension of existing MBRL value gap frameworks to the chunk-level setting, providing useful intuition but limited technical depth. The contribution is best characterized as a well-executed systems integration rather than algorithmic innovation.

2. The UMM backbone (Bagel) is pretrained on interleaved text, image, video, and web data, but has never seen action tokens as an input modality during pretraining. VLA-MBPO introduces discretized robot actions by mapping them onto existing text vocabulary tokens and repurposing their semantics. Finetuning this cross-modal binding with only ~50 trajectories per task raises concerns about whether the model can reliably learn the causal relationship between actions and visual outcomes with so little data. The world model evaluation in Table 1 is limited to in-distribution held-out sets, and no analysis is provided on the robustness of action-conditioned predictions under stronger distribution shift.

3. Neither Table 2 nor Figure 5 reports standard deviations or confidence intervals. Without statistical tests, it is hard to assess whether the differences are significant, especially given the relatively small evaluation budgets.

[1] Park, Kwanyoung, et al. "Scalable Offline Model-Based RL with Action Chunks." *arXiv preprint arXiv:2512.08108* (2025).

---

> ### Author Rebuttal · Authors · 2026-03-31
>
> We sincerely appreciate the reviewer’s constructive feedback and would like to offer further clarification in response.
>
> >**W1**: Clarification of novelty and contributions.
>
> We thank the reviewer for the comment and clarify that our contribution goes beyond a straightforward combination of existing techniques.
>
> First, we address a fundamental and underexplored challenge in world model-based RL for VLA: how to improve VLAs for complex and long-horizon tasks under inevitable compounding model errors. Prior work largely overlooks this issue, while our method is explicitly designed to mitigate it.
>
> Second, **we deliberately chose the simplest and most direct solution to this problem**. For previous model-based RL, simplicity is notoriously hard to maintain without sacrificing performance; most prior work resorts to conservative penalties, separate model ensembles, or task-specific tuning. VLA-MBPO avoids all of these. We believe a principled, minimal framework that works robustly across tasks is itself a meaningful contribution to the community.
>
> Regarding the theoretical contribution, our goal is to provide a rigorous justification for why this specific design is benefit for VLA-MBRL rather than pursuing complexity for its own sake. The analysis bridges the gap between traditional MBRL theory and the unique property of chunk-based VLA control, offering the necessary intuition to explain the empirical success of our chunk-level branched rollout strategy.
>
> >**W2**: Generalizaiton evaluation of UMM-World.
>
> We conduct new OOD generalization evaluations on the high-fidelity RoboTwin benchmark, which provides robust domain randomization capabilities. We evaluated the generation quality of UMM-World across strong distribution shift including unseen backgrounds, textures, objects and lighting conditions.
>
> *(Note: the wrist-view metrics reported below represent the average performance across both the left and right wrist cameras.)*
>
> |Method|LPIPS(H)↓|PSNR(H)↑|SSIM(H)↑|LPIPS(W)↓|PSNR(W)↑|SSIM(W)↑|
> |---|---|---|---|---|---|---|
> |UMM-World|0.12|22.05|0.87|0.25|17.83|0.79|
>
> The results demonstrate that UMM-World maintains strong generative fidelity to severe visual distribution shifts.
>
> >**W3**: Reported standard deviations to assess significance under small budgets.
>
> We update our results with standard deviations across 5 evaluation seeds.The updated performance on the *LIBERO* benchmark is summarized below:
>
> |Model|Spatial|Object|Goal|Long|Average|
> |-|-|-|-|-|-|
> |$\pi_{0.5}$ (SFT) |77.9|89.0|85.2|54.1|76.6|
> |$\pi_{RL}$|85.2±2.1|93.0 ± 1.8|90.2 ± 1.7|61.0 ± 2.5|82.3 ± 2.0|
> |IDQL|79.2±1.6|92.0± 2.2|86.3 ± 1.7|51.9 ± 2.2|77.4 ± 1.6|
> |VLA-MBPO (Ours)|**88.0 ± 2.0**|**95.8 ± 2.2**|**93.7 ± 2.3**|**66.5 ± 1.8**|**86.0 ± 1.9**|
>
> >**Q1**: Clarification of world model training setup and scaling experiments of UMM-World.
>
> In our paper, we indeed followed a suite-specific fine-tuning protocol to maintain experimental consistency. To evaluate the model's scalability, we conduct additional experiments by training a single, unified UMM-World with full data across four task suites. The comparative results on the *LIBERO-Object* evaluation set are summarized below:
>
> |Model|LPIPS(H)↓|PSNR(H)↑|SSIM(H)↑|LPIPS(W)↓|PSNR(W)↑|SSIM(W)↑|
> |-|-|-|-|-|-|-|
> |Suite Specific|0.0940|23.29|0.9060|0.2540|18.76|0.7510|
> |Full Data|**0.0625**|**24.35**|**0.9180**|**0.2149**|**20.05**|**0.7823**|
>
> The results demonstrate that training across multiple suites enhances the world model's performance. The full data model achieves consistently better metrics across all dimensions. This indicates that our framework effectively leverages larger-scale data to learn more robust and accurate dynamics.
>
> >**Q2**: How does performance degrade if the offline trajectories are suboptimal or cover only a narrow portion of the state space?
>
> We clarify that offline datasets used in our experiments are both suboptimal and limited in coverage. Specifically, we use the VLA model after few-shot SFT to interact with the environment and generate the offline dataset, which includes many failure trajectories. Despite these challenging conditions, our method consistently achieves stable and significant improvements in policy performance.
>
> >**Q3**: Reward model evaluation on real-world tasks.
>
> We conduct a dedicated reward evaluation across our diverse real-world tasks. The results are summarized below:
>
> |Metric|Plug-in|Fold Towel|Insert Pen|Pick Cup|Wipe Board|
> |-|-|-|-|-|-|
> |Accuracy|0.994|0.946|0.980|0.988|0.920|
> |F1-Score|0.994|0.957|0.980|0.986|0.931|
>
> Despite the presence of complex backgrounds and varying lighting, the model maintains high accuracy across tasks with different physical properties. The high F1-scores indicate that the reward model can precisely distinguish successful termination states from near-misses or failures, which is critical for providing reliable gradients in world model-based policy optimization.

---

> > ### Author Rebuttal · Reviewer_RgpN · 2026-04-02
> >
> > I appreciate the authors for their rebuttal. The additional experiments adequately address my main concerns. I encourage the authors to further include downstream policy results for the OOD and unified world model settings in the final version. I raise my overall recommendation to Accept.

---

> > > ### Author Response · Authors · 2026-04-02
> > >
> > > We sincerely appreciate your feedback and are glad that the your conecrn have been fully addressed. Thank you for your positive assessment and for raising your overall recommendation to Accept.
> > >
> > > We also thank you for the helpful suggestion regarding the inclusion of downstream policy results for the OOD and unified world model settings. We will incorporate these results in the final version to further strengthen the paper.

---

### Official Review · Reviewer_Hu6C · 2026-03-08

**Soundness:** 2
**Presentation:** 2
**Significance:** 3
**Originality:** 3
**Overall Recommendation:** 4
**Confidence:** 4

**Summary:**

This paper investigates world-model-based RL for VLAs. The proposed VLA-MBPO algorithm extends the general offline model-based RL framework in several aspects: (1) adapting UMMs for joint visual observation and reward prediction; (2) sequential multi-view generation; (3) and chunk-level branched rollout. Empirical evaluations are conducted on the simulated LIBERO benchmark and on real-world robotic systems.

**Compliance With Llm Reviewing Policy:**

Affirmed.

**Final Justification:**

I appreciate the detailed response. In particular, the additional experimental results have largely addressed my concerns. Regarding novelty, I think the practical value of the work—namely, its effective combination of useful techniques—makes it worthy of publication despite the limited novelty.

**Key Questions For Authors:**

1. What is the precise algorithmic description of chunk-level branched rollout? Currently, the paper only provides a citation to Park et al. (2025). It would be helpful to include a clear description in the main text (either in the methods or preliminaries section). The lack of detail also makes it difficult to fully understand the assumptions in Theorem 4.2 and the ablation study in Section 5.4.
2. The method uses PPO as the base RL algorithm. However, PPO is generally effective only for nearly on-policy samples, which appears incompatible with the proposed branched rollouts generated from intermediate states of historical trajectories.
3. "Directly extending UMMs to multi-view inputs often results in view-specific artifacts."  It would be helpful to include qualitative or quantitative results illustrating these artifacts.
4. The value gap in Theorem 4.1 is claimed to "grows quadratically with the task horizon". However, I cannot find any quadratical term regarding task horizon in Eq 6.
5. What do you mean by "Credit Assignment for Stitched Trajectories"? I can only see value prediction of two independent trajectories in Figure 3.

I believe this work has promising potential. If the authors can address the clarity issues and strengthen the empirical comparisons, I would be willing to increase my score.

**Limitations:**

yes

**Strengths And Weaknesses:**

Strengths:

1. This paper focuses on practical model-based RL for VLAs, which is a timely and relevant direction for the community. The proposed algorithmic components are thoughtfully designed and potentially valuable. If clearly presented, this work could have meaningful impact.
2. Using UMMs as world models is an interesting and promising direction.
3. The paper includes a theoretical analysis of value estimation, which is relatively uncommon in this research area and therefore valuable.

Weaknesses:

1. The presentation of the method is not always sufficiently clear. Some algorithmic components are difficult to understand from the current description. See the questions below for details.

2. Lack of baselines:

   - Although the use of UMMs as world models is promising, UMM-World in this paper lacks a comprehensive comparison with recent video world models, which are currently a more widely recognized approach in the community.

   - This paper also lacks comparisons, in either final performance or computational efficiency, with other world-model-based RL methods for VLAs, such as VLA-RFT, World-Env, and WMPO. Currently, the paper only compares with Ctrl-World. This limits the strength of the claims regarding the practical advantages of the proposed approach.

3. The simulated evaluation is conducted only on LIBERO, which has become relatively saturated in the community. Including additional and more challenging benchmarks, such as RoboTwin or RoboCasa, would strengthen the empirical validation.

---

> ### Author Rebuttal · Authors · 2026-03-31
>
> We sincerely appreciate the reviewer’s constructive feedback and would like to offer further clarification in response.
>
> >**W2**: Additional baselines.
>
> We add two recent sota baselines targeting our framework's core components: Cosmos-Predict2.5 [1] for world moedling comparison and WMPO [2] for policy optimization comparison.
>
> **World Modeling Comparison.** Comparison results on LIBERO-Object are shown below. While Cosmos uses massive pre-training to achieve strong general priors in the head-view, it struggles with the wrist-view essential for manipulation.
>
> |Model|LPIPS(H)↓|PSNR(H)↑|SSIM(H)↑|LPIPS(W)↓|PSNR(W)↑|SSIM(W)↑|
> |-|-|-|-|-|-|-|
> |Cosmos-Predict2.5|0.16|**26.84**|**0.91**|0.71|10.76|10.53|
> |UMM-World (Ours)|**0.09**|23.29 |**0.91**|**0.25**|**18.76**|**0.75**|
>
> **Policy Improvement Comparison.** We compare VLA-MBPO against WMPO regarding policy performance and training efficiency. To ensure a fair comparison, we equip WMPO with our UMM-World and Flow-SDE formulation. Even with these enhancements, VLA-MBPO achieves superior success rates across all tasks.
>
> |Method|Spatial|Object|Goal|Long|Average|
> |-|-|-|-|-|-|
> |WMPO|80.6|89.8|85.0|56.6|78.0|
> |VLA-MBPO (Ours)|**87.8**|**96.6**|**92.8**|**66.8**|**85.9**|
>
> Moreover, compred to the original WMPO implementation, our design drastically lowers the barrier to entry for world model training and RL.
>
> |Phase|World Model Training|RL Optimization|
> |-|-|-|
> |WMPO|~4 days on 32 $\times$ H100 GPUs|～1 day|
> |VLA-MBPO (Ours)|**~8 hours on 4 $\times$ H100 GPUs**|**~8 hours**|
>
> >**W3**: Expanded benchmark experiments.
>
> We conduct additional experiments on RoboTwin 2.0 [3]. Specifically, we evaluated VLA-MBPO against the baselines on three representative tasks. The results, summarized in the table below, demonstrate that VLA-MBPO consistently outperforms the base model across all evaluated scenarios.
>
> |Method|Beat Block Hammer|Click Alarmclock|Lift Pot|Average|
> |-|-|-|-|-|
> |$\pi_{0.5}$ (Base)|79|57|59|65|
> |IDQL|78|59|61|66|
> |$\pi_{RL}$|81|61|62|68|
> |**VLA-MBPO (Ours)**|**83**|**65**|**65**|**71**|
>
> >**Q1**: What is the chunk-level branched rollout?
>
> **Branched rollout** refers to policy rollouts that start not from the task’s initial state, but from arbitrary states sampled from the offline dataset. **Chunk-level** means that, during the rollout, the world model directly predicts the outcome of executing a sequence of actions $a_{t:t+k}$ from the current state $s_t$ directly as $s_{t+k+1}$, without generating the intermediate states.
>
> >**Q2**: Clarification of "on-policy" issue in branched rollouts.
>
> We acknowledge the distinction. Strictly speaking, on-policy PPO requires rollouts from the task's initial state distribution, our branched rollouts relax this by starting from intermediate states. But once a branch begins, all subsequent interactions are fully on-policy. This relaxation is well-precedented (e.g., Dreamer-v3 [4] also applies on-policy Policy Gradient on branch rollout setting) and our experiments confirm it remains effective in practice.
>
> >**Q3**: Included qualitative results demonstrating view-specific artifacts in multi-view inputs.
>
> We thank the reviewer for this constructive suggestion. We have provided qualitative examples at [artifacts](https://anonymous.4open.science/r/dascdjaskldngalser/artifacts.pdf). As shown, directly extending UMMs causes severe cross-view spatial inconsistencies, such as conflicting object states between the head and wrist cameras. We will include these visual comparisons in the revised appendix.
>
> >**Q4**: Clarification of  "grows quadratically with the task horizon" in Theorem 4.1.
>
> This refer to the effective horizon $H\approx 1/1-\gamma$. From Eq. (6), with constant chunk size $k$, the value gap is scale as $\mathcal{O}(\frac{1}{(1-\gamma)(1-\gamma^k)})\approx \mathcal{O}(\frac{1}{k(1-\gamma)^2})=\mathcal{O}(H^2)$. Thus, the quadratic growth is w.r.t. the effective horizon.
>
> >**Q5**: Clarification of "Credit Assignment for Stitched Trajectories".
>
> We apologize for the confusion. During training, we only rollout two chunks from each sampled point (a snippet of trajectory) for value and policy learning. When evaluation, we test on whole trajectories composed of more than 10 chunks. Thus, “stitched trajectories” refer to this test-time composition: the value at the end of one chunk must accurately reflect the expected reward over all subsequent chunks, even though no such long-horizon signal was directly observed during training.
>
> **References**
>
> [1] Cosmos world foundation model platform for physical ai. arXiv preprint arXiv:2501.03575, 2025.
>
> [2] WMPO: World Model-based Policy Optimization for Vision-Language-Action Models. ICLR'26.
>
> [3] RoboTwin 2.0: A scalable data generator and benchmark with strong domain randomization for robust bimanual robotic manipulation. arXiv preprint arXiv:2506.18088.
>
> [4] Mastering Diverse Domains through World Models. arXiv preprint arXiv:2301.04104 (2023).

---

> > ### Author Rebuttal · Reviewer_Hu6C · 2026-04-02
> >
> > I appreciate the detailed response. In particular, the additional experimental results have largely addressed my concerns. Regarding novelty, I think the practical value of the work—namely, its effective combination of useful techniques—makes it worthy of publication despite the limited novelty.

---

> > > ### Author Response · Authors · 2026-04-02
> > >
> > > We are happy that all your concerns have been addressed. We sincerely appreciate the reviewer for raising the score.

---

### Official Review · Reviewer_qEwp · 2026-03-11

**Soundness:** 2
**Presentation:** 3
**Significance:** 3
**Originality:** 2
**Overall Recommendation:** 3
**Confidence:** 3

**Summary:**

This paper introduces VLA-MBPO, a framework that utilizes a Unified Multimodal Model (UMM) as a world model to improve the efficiency and accuracy of VLA fine-tuning, offering an alternative to computationally expensive video generation backbones. The method fine-tunes a UMM to accept low-level action chunks as additional inputs. To ensure consistency across multi-view camera observations, the authors propose "Interleaved View Decoding," where the transition function generates the head view first and subsequently generates the wrist view conditioned on the predicted head view.

For RL fine-tuning, the authors employ a chunk-level branched rollout scheme that initiates imaginary sub-trajectories from arbitrary states sampled from the offline dataset. This approach aims to mitigate accumulated model error while maintaining sufficient data diversity for policy improvement. Theoretical analysis is provided to show that this rollout scheme tightens the bound on the value gap. Experiments demonstrate: 1) the world model's performance against baselines, 2) the improvement of the RL fine-tuned policy over supervised baselines in both simulated and real-world environments, and 3) the sensitivity of the method to hyperparameters such as branch length and sample size.

**Compliance With Llm Reviewing Policy:**

Affirmed.

**Final Justification:**

The rebuttal has addressed my concerns. I'll maintain my positive score.

**Key Questions For Authors:**

1. Regarding IVD: Table 1 compares the proposed method against a baseline without IVD, which I assume utilizes parallel generation. Did the authors experiment with the reverse decoding order (generating the wrist view first, then the head view) to empirically verify the claim that "global information" from the head view is the requisite prior for consistent wrist view generation?

2. RL Hyperparameters: How many RL iterations ($N_{RL, iter}$) were performed for the reported VLA-MBPO results? I could not find a specific value for the number of iterations or the total interaction budget in the hyperparameters section or Algorithm 1.

3. Baselines: Given the rapid developments in this field, did the authors consider comparing the world model's performance against NVIDIA Cosmos?

4. Theoretical Clarification: In Section 4.1, the text states that "the value gap grows quadratically with the task horizon due to the compounding errors". However, Equation 6 is an upper bound involving the discount factor $\gamma$. Could the authors clarify if this "quadratic growth" refers to the effective horizon, $H \approx 1/(1-\gamma)$?

**Limitations:**

yes

**Strengths And Weaknesses:**

# Strengths

- Relevance and Efficiency: The paper addresses a significant bottleneck in the reinforcement learning fine-tuning of VLA models by leveraging world models that are less computationally intensive than standard video generation models.
- Empirical Performance: The policies fine-tuned using VLA-MBPO demonstrate strong performance improvements compared to baseline policies in both simulated benchmarks and real-world tasks.
- Justification of Components: The core design choices—specifically Interleaved View Decoding and chunk-level branched rollouts—are supported by both empirical ablation studies and theoretical analysis.

# Weaknesses

- Justification for Decoding Order: While the ablation study in Table 1 effectively demonstrates that Interleaved View Decoding (IVD) outperforms parallel generation, the reasoning behind the specific order (Head $\to$ Wrist) is not sufficiently explored. The paper argues that global information is needed for local details, but it does not empirically compare this against the reverse order (Wrist $\to$ Head) or other conditional dependencies to validate this claim.
- Novelty of Rollout Scheme: The proposal to branch imaginary rollouts from random states in the dataset is presented as a contribution. However, similar "foresight" approaches that collect imaginary rollouts starting from visited or dataset states have been explored in prior work, such as MapGo (Zhu et al., MapGo: Model-Assisted Policy Optimization for Goal-Oriented Tasks, 2021). The paper would benefit from clarifying how the proposed chunk-level branching differs from or improves upon these existing foresight methods.

---

> ### Author Rebuttal · Authors · 2026-03-31
>
> We sincerely appreciate the reviewer’s constructive feedback and would like to offer further clarification in response.
>
> > **W1 & Q1**: Ablation experiments on view decoding.
>
> To verify our design choice of interleaved view decoding, we conducted an ablation study by reversing the decoding order (**Wrist-first**) and compared it with our proposed (**Head-first**) order. The quantitative results for generative quality on *LIBERO-Object* are summarized below:
> |Decoding Order|LPIPS(H)↓|PSNR(H)↑|SSIM(H)↑|LPIPS(W)↓|PSNR(W)↑|SSIM(W)↑|
> |-|-|-|-|-|-|-|
> |Wrist-first|0.096|23.12|**0.910**|0.393|14.84|0.626|
> |Head-first (Ours)|**0.094**|**23.29**|0.906|**0.254**| **18.76**|**0.751**|
>
> As shown in the table, while both decoding orders yield comparable performance in head-view generation, our strategy leads to a substantial improvement in wrist-view quality compared to the Wrist-first alternative. This disparity empirically confirms our hypothesis: the head-view provides essential macro-dynamics and global spatial context. By using this global information as a prior, the model can more accurately ground the local details of the wrist-view, effectively reducing visual ambiguity and ensuring cross-view consistency.
>
> > **W2**: Novelty of Rollout Scheme.
>
> We thank the reviewer for the insightful comment. While both our method and prior work such as MapGo leverage model-based rollouts from dataset states, they differ fundamentally in how model error is handled and the level of rollouts.
>
> MapGo [1] adopts a conservative design: due to model bias, it limits step-level rollouts to short horizons and requires continual correction from real-environment samples during training, resulting in inherently local, near-future foresight. In contrast, our method overcomes these limitations through three key design choices:
>
> 1. **Chunk-level branching**: We generate and reuse temporally extended trajectory segments, enabling longer horizons with fewer interaction steps while maintaining temporal coherence.
> 2. **Pretrained world model**: This enhances generalization in the small data regime, supporting longer-horizon rollouts without severe model error accumulation.
> 3. **Purely offline for RL**: Once prior data is available, the policy can learn entirely from the world model, removing the need for further real-environment interaction and making deployment more scalable.
>
> Beyond rollouts, our whole method design deliberately adopts a minimalist architecture by utilizing a unified model, native PPO updates, and shared hyperparameters across tasks without any complex conservative penalty. **Achieving such simplicity without sacrificing performance is itself non-trivial in model-based RL and is key to robustness and scalability.**
>
> We will clarify these distinctions and discuss the connection to MapGo in the revised version.
>
> > **Q2**: Clarification of RL iterations ($N_{RL, iter}$).
>
> We apologize for this omission. For all reported results, we performed a total of **$N = 10$** RL iterations. We will clarify it in the revised manuscript.
>
> > **Q3**: Additional world modeling baseline.
>
> We conduct a comparative evaluation with Cosmos-Predict2.5 [2] on the LIBERO-Object benchmark. The results are summarized below:
>
> |Model|LPIPS(H)↓|PSNR(H)↑|SSIM(H)↑|LPIPS(W)↓|PSNR(W)↑|SSIM(W)↑|
> |-|-|-|-|-|-|-|
> |Cosmos-Predict2.5|0.16|**26.84**|**0.91**|0.71|10.76|10.53|
> |UMM-World (Ours)|**0.09**|23.29 |**0.91**|**0.25**|**18.76**|**0.75**|
>
> While Cosmos demonstrates competitive performance in the head-view, this strength is largely attributed to its massive pre-training on diverse video and action datasets, which equips it with strong general priors for scene reconstruction. Despite its strong head-view performance, Cosmos struggles significantly with the wrist-view, where it's notably inferior to our UMM-World.
>
> >**Q4**: Clarification of "quadratic growth" and effective horizon $H$.
>
> By "quadratic growth" we mean growth with respect to the effective horizon $H\approx 1/1-\gamma$. From Eq. (6), with constant chunk size $k$, the value gap is scale as $\mathcal{O}(\frac{1}{(1-\gamma)(1-\gamma^k)})\approx \mathcal{O}(\frac{1}{k(1-\gamma)^2})=\mathcal{O}(H^2)$. Hence, the term "quadratic growth" refers to the fact that the value gap increases quadratically with the effective horizon.
>
> **References:**
>
> [1] MapGo: Model-Assisted Policy Optimization for Goal-Oriented Tasks. IJCAI'21.
>
> [2] Cosmos world foundation model platform for physical ai. arXiv preprint arXiv:2501.03575, 2025.

---

> > ### Author Rebuttal · Reviewer_qEwp · 2026-04-05
> >
> > I appreciate the author's response.
> > The additional clarifications adequately address my concerns.

---

> > > ### Author Response · Authors · 2026-04-05
> > >
> > > Thank you for your acknowledgment and for taking the time to review our responses. We are glad that your concerns have been fully addressed.
> > >
> > > We appreciate your positive evaluation and would be grateful if you would consider raising your score accordingly. Thank you again for your time and feedback.

---

### Official Review · Reviewer_bmqp · 2026-03-13

**Soundness:** 2
**Presentation:** 3
**Significance:** 3
**Originality:** 3
**Overall Recommendation:** 4
**Confidence:** 4

**Summary:**

In this paper, authors propose VLA-MBPO, which is a world model-based reinforcement learning framework for vision-language-action models. VLA-MBPO has an interleaved view decoding mechanism to enforce multi-view consistency and a chunk-level branched rollout to mitigate error compounding. Experiments on simulation and real-world tasks demonstrate the effectiveness of VLA-MBPO.

**Compliance With Llm Reviewing Policy:**

Affirmed.

**Final Justification:**

While most of my concerns have been resolved, after reading the other reviewers' comments, the original technical novelty is rather weak. But considering the sufficient experimental verification, I finally decided to raise my score.

**Key Questions For Authors:**

See Weaknesses.

**Limitations:**

Yes

**Strengths And Weaknesses:**

Strength:
- The paper addresses an important and timely problem, i.e., how to improve sample efficiency for vision-language-action models using model-based reinforcement learning.
- Experimental results show performance gains and improved sample efficiency across multiple tasks compared to baselines.
- The paper validates the method not only in simulation but also on real robot tasks, which strengthens the practical relevance of the work.

Weakness
- The proposed VLA-MBPO framework mainly integrates several existing techniques from model-based reinforcement learning and VLA models. While the combination is interesting, the core algorithmic contribution appears incremental.
- The experiments compare against a limited number of baselines. Including stronger and more recent baselines would make the empirical evaluation more convincing.
- The paper focuses on demonstrating performance improvements but provides little analysis of the world model’s failure cases. A deeper investigation of prediction errors and rollout degradation would strengthen the paper.
- Although real robot experiments are included, the number of tasks and environments remains limited. It is therefore unclear how well the method generalizes to more complex or diverse real-world scenarios.
- Theoretical results provide bounds under certain assumptions, but their connection to practical implementation is not well explained. For example, it is unclear how the theory informs key design choices such as rollout length or model usage.
- Computational cost is not well discussed. More analysis of training cost and scalability would be helpful.

---

> ### Author Rebuttal · Authors · 2026-03-31
>
> We sincerely appreciate the reviewer’s constructive feedback and would like to offer further clarification in response.
>
> >**W1**: Novelty and technical contributions.
>
> We thank the reviewer for the comment and clarify that our contribution goes beyond a straightforward combination of existing techniques.
>
> First, we address a fundamental and underexplored challenge in world model-based RL for VLA: how to improve VLAs for complex and long-horizon tasks under inevitable compounding model errors. Prior work largely overlooks this issue, while our method is explicitly designed to mitigate it.
>
> Second, **we deliberately chose the simplest and most direct solution to this problem**. For previous model-based RL, simplicity is notoriously hard to maintain without sacrificing performance; most prior work resorts to conservative penalties, separate model ensembles, or task-specific tuning. VLA-MBPO avoids all of these while remaining theoretically grounded. We believe a principled, minimal framework that works robustly across tasks is itself a meaningful contribution to the community.
>
> >**W2**: Stronger and more recent baselines.
>
> We add two recent sota baselines targeting our framework's core components: Cosmos-Predict2.5 [1] for world moedling comparison and WMPO [2] for policy optimization comparison.
>
> **World Modeling Comparison.** Comparison results on LIBERO-Object are shown below. While Cosmos uses massive pre-training to achieve strong general priors in the head-view, it struggles with the wrist-view essential for manipulation.
>
> |Model|LPIPS(H)↓|PSNR(H)↑|SSIM(H)↑|LPIPS(W)↓|PSNR(W)↑|SSIM(W)↑|
> |-|-|-|-|-|-|-|
> |Cosmos-Predict2.5|0.16|**26.84**|**0.91**|0.71|10.76|10.53|
> |UMM-World (Ours)|**0.09**|23.29 |**0.91**|**0.25**|**18.76**|**0.75**|
>
> **Policy Improvement Comparison.** We compare VLA-MBPO against WMPO regarding policy performance and training efficiency. To ensure a fair comparison, we equip WMPO with our UMM-World and Flow-SDE formulation. Even with these enhancements, VLA-MBPO achieves superior success rates across all tasks.
>
> |Method|Spatial|Object|Goal|Long|Average|
> |-|-|-|-|-|-|
> |WMPO|80.6|89.8|85.0|56.6|78.0|
> |VLA-MBPO (Ours)|**87.8**|**96.6**|**92.8**|**66.8**|**85.9**|
>
> Moreover, compred to the original WMPO implementation, our chunk-level branched rollout design drastically lowers the barrier to entry for world model training and RL.
>
> |Phase|World Model Training|RL Optimization|
> |-|-|-|
> |WMPO|~4 days on 32 $\times$ H100 GPUs|～1 day|
> |VLA-MBPO (Ours)|**~8 hours on 4 $\times$ H100 GPUs**|**~8 hours**|
>
> >**W3**: Deeper failure case study.
>
> While Appendix E.2 already provides some failure case analysis, we will add deeper investigations in the revised manuscript as suggested. Specifically, we will detail prediction failures involving complex deformable objects, where the model struggles to simulate infinite-DOF shape changes, leading to unnatural morphing. We will also analyze contact-rich manipulations, where severe gripper occlusions and precise physics often result in blurred contact surfaces or physically implausible penetrations.
>
> >**W4**: More complex real-world experiments.
>
> We expand our real-world experiments to more complex tasks, as seen in [objects](https://anonymous.4open.science/r/asdnvdasretvcda/objects.pdf). The extended evaluation covers **50 diverse subtask combinations**, supported by a dataset of **200 trajectories**.
>
> Our testing protocol, totaling **100 evaluation trials**, is specifically designed to test generalization across three perspectives, including object, lightning and background.
>
> As shown in the table below, VLA-MBPO consistently outperforms all baselines. The results confirm that our world model-based approach enables the policy to better handle OOD visual and physical variations.
>
> ||$\pi_{0.5}$|IDQL|VLA-MBPO|
> |-|-|-|-|
> |Success Rate|36|44|**51**|
>
> > **W5**: The link between the theoretical results and practical design choices.
>
> Our theory directly informs design choices by motivating both a **chunk-level dynamics model** and **chunk-level branched rollouts**. Specifically, the former reduces the policy error from $\gamma^{k}$ to $\gamma^{kn}$, while the latter further decreases the model error term from $\mathcal{O}\left(\frac{k\gamma^k}{(1-\gamma)(1-\gamma^k)}\right)$ to $\mathcal{O}\left(\frac{n}{1-\gamma}\right)$. It also reveals a trade-off in the rollout horizon $n$: larger $n$ reduces policy error but increases model error. Guided by this, we set $n=2$, which performs consistently well across simulated and real-world tasks.
>
> > **W6**: More detailed information of computational cost.
>
> Please see Response to W2. It can be observed that our method requires much less resources compared to existing VLA-MBRL method.
>
> **References:**
>
> [1] Cosmos world foundation model platform for physical ai. arXiv preprint arXiv:2501.03575, 2025.
>
> [2] WMPO: World Model-based Policy Optimization for Vision-Language-Action Models. ICLR'26.

---

> > ### Author Rebuttal · Reviewer_bmqp · 2026-04-04
> >
> > Thank you to the authors for their detailed responses. While most of my concerns have been resolved, after reading the other reviewers' comments, the original technical novelty is rather weak. But considering the sufficient experimental verification, I finally decided to raise my score.

---

> > > ### Author Response · Authors · 2026-04-05
> > >
> > > Thank you for taking the time to carefully consider both our responses and the broader discussion. We are pleased that your concerns have been fully resolved and appreciate your decision to raise the score.
> > >
> > > We notice that the updated score may not yet be reflected in the system. We would greatly appreciate it if you could confirm or update it at your convenience.
> > >
> > > Thank you again for your time and consideration.

---

### Decision · Program_Chairs · 2026-04-30

**Decision:**

Accept (regular)

**Comment:**

This work introduces VLA-MBPO, a model-based approach for robot control. The core idea is to adapt VLMs to robot actions through interleaved view decoding for cross-view consistency and chunk-level rollouts to reduce error accumulation. The authors demonstrate the utility of the proposed method through experiments on both simulated and real robots.

Some reviewers raised concerns regarding baselines and requested additional experiments, but most of these were adequately addressed during the rebuttal. I believe this work makes a valuable contribution to the community and therefore recommend weak accept.